# Generative Augmented Inference

Cheng Lu [1]   Mengxin Wang [2]   Dennis J. Zhang [1]   Heng Zhang [3]

## Abstract

Large language models enable inexpensive AI-generated annotations, but using them reliably for causal inference remains challenging. Naively pooling AI and human data induces bias, while existing methods such as Prediction-Powered Inference (PPI; Angelopoulos et al., 2023a) treat AI outputs as proxies of true labels – an assumption often violated for generative model outputs in practice. We propose Generative Augmented Inference (GAI), a framework that treats AI outputs as general, potentially high-dimensional informative features for learning human labels rather than as surrogates. GAI flexibly models this relationship using nonparametric methods, enabling consistent estimation and valid inference from combined human and AI data. We establish asymptotic normality and show that, under random labeling, GAI strictly improves asymptotic efficiency over human-data-only estimation whenever AI outputs are informative for true labels. Empirical studies on real-world datasets demonstrate that GAI significantly reduces estimation error and improves confidence interval quality across diverse generative data sources relative to human-only and PPI-based estimation.

## 1. Introduction

The emergence of large language models (LLMs) has fundamentally transformed the economics of data collection. Models like GPT-4 can annotate text, answer surveys, and produce labels at costs orders of magnitude lower than human annotation (Brown et al., 2020; OpenAI, 2023). This capability is transformative across empirical research: conjoint analysis in marketing ($5–20 per respondent), medical image annotation ($50–500 per case), sentiment analysis for social science ($0.50–2 per document), and legal document review ($25–150 per hour). When LLM API calls cost $0.01–0.10 per annotation, the potential for $10$–$1000\times$ cost reduction is enormous. This shift has the potential to transform empirical research by enabling orders-of-magnitude increases in sample size at negligible marginal cost.

However, realizing these efficiency gains while maintaining valid statistical inference presents a fundamental challenge. AI representations can differ systematically from human judgments in complex, feature-dependent ways that are difficult to characterize. Existing AI-augmented inference methods implicitly assume AI outputs are noisy surrogates of the target variable. This assumption is fundamentally incompatible with modern generative models, whose outputs are often categorical, biased, high-dimensional, or uncalibrated. Treating AI outputs as interchangeable with human labels—or even as noisy versions of them—can introduce bias that overwhelms any efficiency gains.

GAI replaces the surrogate-label view with a feature-augmentation view: AI outputs are treated as arbitrary side information that helps predict human labels, not as substitutes for them. This single shift unlocks valid inference even when AI outputs are inaccurate, discrete, or systematically biased. The central question is therefore not whether AI representations are "accurate enough," but rather: *How AI-generated signals can be incorporated into estimation without requiring them to serve as surrogate labels for the outcome of interest?*

**Existing Approaches.** A prominent line of work, *Prediction-Powered Inference* (PPI; Angelopoulos et al., 2023a;b), leverages predictions on unlabeled data to reduce uncertainty by correcting prediction bias using a labeled subsample. PPI is highly effective when predictions are well-calibrated and closely approximate the outcome. However, its variance reduction hinges on prediction quality: when predictions are discrete, moderately accurate, or systematically biased, the correction term can amplify noise rather than reduce it. In one of our experiments with LLM-generated labels achieving 54% accuracy, PPI performs substantially worse than both primary-only estimation and our proposed method.

[1]Olin Business School, Washington University in St. Louis, USA [2]Naveen Jindal School of Management, University of Texas at Dallas, USA [3]W. P. Carey School of Business, Arizona State University, USA. Correspondence to: Mengxin Wang <mengxin.wang@utdallas.edu>.

*Proceedings of the 43rd International Conference on Machine Learning*, Seoul, South Korea. PMLR 306, 2026. Copyright 2026 by the author(s).

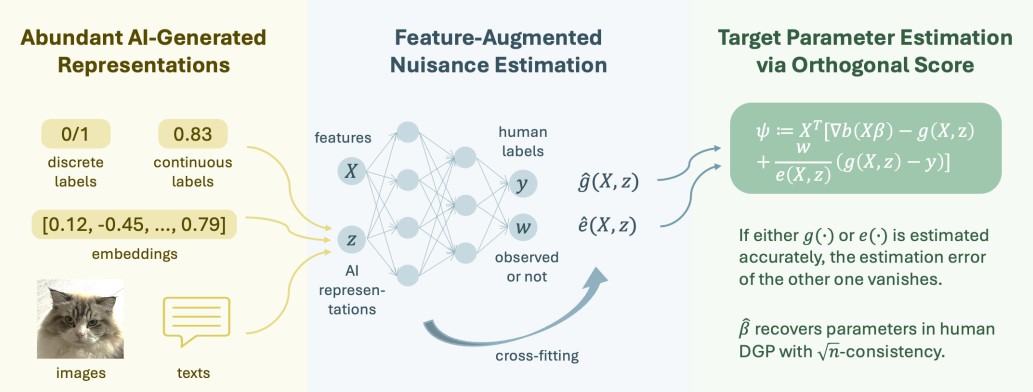

*Figure 1.* GAI incorporates AI outputs as auxiliary features rather than surrogate labels. In contrast to PPI, which applies corrections at the loss-function level and relies on AI predictions approximating the outcome, GAI embeds auxiliary signals within a Neyman-orthogonal score function. This construction allows the estimator to correct bias and improve efficiency by leveraging auxiliary data, even when AI representations are discrete, biased, or weakly informative.

**Our Approach.** We propose a fundamentally different perspective grounded in Neyman orthogonality (Chernozhukov et al., 2018). Rather than treating AI representations as proxies for the outcome, we treat them as *general auxiliary informative features* that help predict the outcome. Crucially, AI representations are *not required* to resemble human labels in scale, support, or interpretation. It may be categorical, biased, high-dimensional, or unstructured. Its role is purely predictive, not substitutive. This distinction fundamentally separates our framework from PPI and related approaches.

By embedding AI representations into a Neyman-orthogonal score, we construct an estimator that (i) remains computable when human labels are missing, (ii) is robust to flexible, nonparametric estimation of nuisance components, such as using machine learning (ML), and (iii) admits valid $\sqrt{n}$-inference under standard Neyman orthogonality conditions.

**Contributions.** Our contributions are threefold:

- We develop GAI for AI-augmented estimation in generalized linear models (GLMs) that accommodates partially observed outcomes and treats AI representations as flexible covariates rather than surrogate labels (Section 4).
- We establish asymptotic normality and prove that our estimator strictly dominates primary-only estimation in asymptotic variance under random selection whenever AI representations are informative (Theorem 5.3, Corollary 5.4).
- We demonstrate significant empirical gains in two contrasting auxiliary-data regimes—noisy, discrete LLM representations in conjoint analysis and well-calibrated continuous representations in census data—where GAI consistently outperforms PPI-based methods in estimation accuracy and achieves valid coverage without relying on inflated confidence intervals (Section 6).

**Paper Organization.** Section 2 reviews related work. Section 3 formalizes the setting. Section 4 presents the methodology. Section 5 establishes theoretical guarantees. Section 6 reports empirical results. Section 7 concludes.

## 2. Related Work

**Semi-Supervised and Transfer Learning.** Classical semi-supervised methods (Nigam et al., 2000; Zhu & Goldberg, 2009; Lee et al., 2013) use unlabeled features to improve prediction. Transfer learning (Pan & Yang, 2009; Weiss et al., 2016) addresses distribution shift between domains. Our setting is distinct: we have AI-generated *representations* (not just labels) for auxiliary data, but these data may be systematically biased. The AI-augmented estimator (AAE) framework (Wang et al., 2024) addresses this through parametric debiasing. Unlike AAE, GAI is fully nonparametric and remains valid under heterogeneous, high-dimensional, or unstructured AI representations.

**Missing Data.** Our framework connects to missing data theory (Rubin, 1976; Little & Rubin, 2019), where human labels are "missing" for auxiliary observations. The inverse probability weighting term in our score function mirrors classical IPW estimators (Hirano et al., 2003), while the weighted residual term provides augmentation analogous to AIPW (Glynn & Quinn, 2010). The key distinction is that we observe AI representations for all observations, providing rich auxiliary information beyond the missingness indicator.

**Prediction-Powered Inference.** Angelopoulos et al. (2023a) introduced PPI for constructing valid confidence intervals by leveraging ML predictions on unlabeled data. PPI uses labeled data to estimate and correct for the bias in predictions, yielding valid inference when predictions are reasonably accurate proxies for outcomes. When pre-

dictions are continuous and well-calibrated, PPI achieves substantial efficiency gains. However, PPI's variance reduction depends on prediction quality: the correction term has variance that grows when predictions poorly approximate outcomes. In some cases, AI outputs are categorical rather than continuous probability estimates, violating the implicit assumption that AI-generated data and human labels live in the same space. Our framework instead treats AI representations as *features* for predicting outcomes, which remains valid regardless of AI accuracy.

**Neyman orthogonality and Doubly Robust Estimation.** Chernozhukov et al. (2018) developed double machine learning for causal inference with high-dimensional nuisance parameters, building on the doubly robust estimation literature (Bang & Robins, 2005). The Neyman orthogonality condition ensures that estimation error in nuisance functions has only second-order effects on target parameters. Our score function (Equation 3) shares the doubly robust structure: it remains consistent if one of the nuisance functions is correctly specified, though we require both for efficiency. Kennedy (2022) provides a comprehensive review of these connections.

## 3. Problem Formulation

### 3.1. Setting and Notation

Consider a dataset $\mathcal{D} = \{\Xi_i = (\mathbf{X}_i, \mathbf{y}_i, w_i, \mathbf{z}_i)\}_{i=1}^n$ of i.i.d. observations where:

- $\mathbf{X} \in \mathcal{X} \subset \mathbb{R}^{k \times d}$: feature matrix
- $\mathbf{y} \in \mathbb{R}^k$: human label of interest (partially observed)
- $w \in \{0, 1\}$: indicator for whether human label is observed
- $\mathbf{z}$ is AI-generated auxiliary information, whose structure is left deliberately unrestricted. $\mathbf{z}$ may be structured (e.g., categories, probabilities), high-dimensional vectors (e.g., embeddings), or unstructured objects (e.g., texts, images).

Define the *primary* (human-labeled) and *auxiliary* (AI-only) subsamples $\mathcal{D}^{\mathrm{P}} = \{\Xi_i \in \mathcal{D} : w_i = 1\}$, $\mathcal{D}^{\mathrm{A}} = \{\Xi_i \in \mathcal{D} : w_i = 0\}$ and let $n_{\mathrm{P}} = |\mathcal{D}^{\mathrm{P}}|$, $n_{\mathrm{A}} = |\mathcal{D}^{\mathrm{A}}|$.

### 3.2. Generalized Linear Models and Applications

Let $b(\cdot) : \Theta \to \mathbb{R}$ be a known convex function on open convex $\Theta \subset \mathbb{R}^k$. Define $\ell(\mathbf{X}, \mathbf{y}; \boldsymbol{\beta}) := b(\mathbf{X}\boldsymbol{\beta}) - \mathbf{y}^\top \mathbf{X}\boldsymbol{\beta}$. The target parameter $\boldsymbol{\beta}^* \in \mathcal{B} \subset \mathbb{R}^d$ solves:

$$\boldsymbol{\beta}^* \in \arg\min_{\boldsymbol{\beta} \in \mathcal{B}} \mathbb{E}\left[\ell(\mathbf{X}, \mathbf{y}; \boldsymbol{\beta})\right], \quad (1)$$

satisfying the first-order condition:

$$\mathrm{E}[\nabla_{\boldsymbol{\beta}} \ell(\mathbf{X}, \mathbf{y}; \boldsymbol{\beta}^*)] = \mathbb{E}\left[\mathbf{X}^\top \left(\nabla b(\mathbf{X}\boldsymbol{\beta}^*) - \mathbf{y}\right)\right] = 0 \quad (2)$$

We assume that $\mathcal{B}$ is an open convex set such that, for some compact set $\breve{\mathcal{X}}$ with $\mathcal{X} \subset \breve{\mathcal{X}}$, it holds that $\mathbf{X}\boldsymbol{\beta} \in \Theta$ whenever $\mathbf{X} \in \breve{\mathcal{X}}$ and $\boldsymbol{\beta} \in \mathcal{B}$.

This framework covers canonical GLMs that appear throughout applied modeling pipelines in both industry and scientific applications:

- *Linear regression:* $k = 1$, $b(\theta) = \frac{1}{2}\theta^2$. This corresponds to squared-loss estimation for continuous outcomes and is the workhorse model in empirical studies, including treatment-effect estimation, forecasting, and policy evaluation. Note that linear regression, viewed as special case of GLM, requires normal densities conditional on $\mathbf{X}$.
- *Multinomial logit (MNL):* for $k$ non-baseline classes, $b(\boldsymbol{\theta}) = \log\left(1 + \sum_{j=1}^k \exp(\theta_j)\right)$. This is the standard model for discrete choice among multiple alternatives, which is widely used for multiclass classification applications, such as text classification, image recognition, and recommendation systems, where predicted utilities determine selection probabilities among competing options. When $k = 1$, this reduces to logistic regression.
- *Poisson regression:* $k = 1$, $b(\theta) = \exp(\theta)$. This models count outcomes and event rates, with applications such as demand incidence, click/conversion counts, and arrival processes in computer systems. Poisson GLMs are standard for modeling event rates in systems monitoring, reliability, and online experimentation.

Canonical GLMs yield score equations with convex objective functions and thus fits well into our framework. Under canonical GLM, (1) is interpreted as the expected log-likelihood, and $\exp\left\{b(\mathbf{X}\boldsymbol{\beta}^*) - \mathbf{y}^\top \mathbf{X}\boldsymbol{\beta}^*\right\} \propto f(\mathbf{y} \mid \mathbf{X})$, i.e., the true density. However, critically, our approach allows the model to be misspecified. For instance, with a $b(\cdot)$ function from GLM family, we are seeking a GLM density that minimizes the conditional Kullback–Leibler (KL) divergence. One example is that in the MNL setting, the true choice model can be different from the MNL model. Furthermore, for linear regression where $\nabla b(\mathbf{X}\boldsymbol{\beta}) = \mathbf{X}\boldsymbol{\beta}$, our framework does not require normal density, or even $\mathbf{X}\boldsymbol{\beta}$ being the correctly specified conditional mean. In this case, (2) gives the best linear approximation to the conditional mean. Therefore, we broadly view (1) and (2) as a *best-in-class* estimator, which can either be correctly or incorrectly specified.

### 3.3. Selection Mechanism

We assume human labels are unobserved at random: $w \perp \mathbf{y} \mid \mathbf{X}, \mathbf{z}$. The conditional independence assumption is natural when primary data collection is designed (e.g., random sampling of human annotations). Define:

- $e^*(\mathbf{X}, \mathbf{z}) = \mathbb{E}[w \mid \mathbf{X}, \mathbf{z}] > \kappa$ for some $\kappa > 0$
- $\mathbf{g}^*(\mathbf{X}, \mathbf{z}) = \mathbb{E}[\mathbf{y} \mid \mathbf{X}, \mathbf{z}]$

Unless noted otherwise, we let $\|\cdot\|$ denote the $\ell_2$-norm for a matrix or a vector. Also, $\|\cdot\|_F$ and $\|\cdot\|_\infty$ are the Frobenius norm or the $\ell_\infty$-norm of an appropriate object. Fur-

thermore, given a multi-dimensional function $\mathbf{f}$, we write $\|\mathbf{f}\|_{P,2} = \left(\mathrm{E}[\|\mathbf{f}\|^2]\right)^{1/2}$. We use $\lambda_{\min}(\cdot)$ to denote the minimum eigenvalue of a matrix.

# 4. Methodology

## 4.1. The Challenge

The canonical score $\mathbf{X}^\top(\nabla b(\mathbf{X}\boldsymbol{\beta}) - \mathbf{y})$ in (2) requires observing $\mathbf{y}$, so it cannot be evaluated on auxiliary observations with $w = 0$. Our goal is to construct a score function that:

1. identifies $\boldsymbol{\beta}^*$ via a valid population moment condition;
2. is computable for *all* observations, including those with missing $\mathbf{y}$;
3. is *Neyman-orthogonal* with respect to nuisance functions $(e, \mathbf{g})$, yielding valid inference under flexible ML estimation.

## 4.2. The Score Function

We propose the score function:

$$\boldsymbol{\psi}(\Xi; e, \mathbf{g}; \boldsymbol{\beta}) :=$$
$$\mathbf{X}^\top\left[\nabla b(\mathbf{X}\boldsymbol{\beta}) - \mathbf{g}(\mathbf{X}, \mathbf{z}) + \frac{w}{e(\mathbf{X}, \mathbf{z})}(\mathbf{g}(\mathbf{X}, \mathbf{z}) - \mathbf{y})\right] \quad (3)$$

This proposed score can be viewed as an *orthogonalized* version of the complete-data score: it replaces the missing label $\mathbf{y}$ by the regression function $\mathbf{g}(\mathbf{X}, \mathbf{z})$ and corrects the residual using IPW through $e(\mathbf{X}, \mathbf{z})$. This construction allows auxiliary data (with $w = 0$) to contribute information through $\mathbf{g}(\mathbf{X}, \mathbf{z})$, even though $\mathbf{y}$ is unobserved. For brevity, we sometimes suppress the dependency on $\mathbf{X}$ and $\mathbf{z}$ in $e(\cdot)$ and $\mathbf{g}(\cdot)$ when clear from the context.

## 4.3. The Algorithm

We now propose the GAI algorithm used to estimate the target parameters $\boldsymbol{\beta}^*$. Algorithm 1 summarizes the full procedure; in practice, it reduces to fitting two ML models followed by a single convex optimization.

# 5. Theoretical Results

Before stating formal asymptotic normality, we highlight a key efficiency result: under random labeling, GAI strictly dominates primary-only estimation whenever AI outputs are informative, even when they are inaccurate. These nice properties are built upon the following assumptions:

**Assumption 5.1** (Regularity). (i) $\mathbf{X} \in \mathcal{X}$ is bounded; (ii) $b(\cdot)$ is twice continuously differentiable with $\nabla^2 b(\theta) \succ 0$ on $\Theta$; (iii) $\mathbb{E}[\mathbf{X}^\top\mathbf{X}]$ is positive definite; (iv) $\|\mathrm{Cov}(\mathbf{y} \mid \mathbf{X}, \mathbf{z})\| \leqslant \tilde{\sigma}^2$.

---

**Algorithm 1** Generative Augmented Inference

**Input:** Data $\mathcal{D} = \mathcal{D}^P \cup \mathcal{D}^A$, number of folds $K$

1: Randomly partition all data $\mathcal{D}$ into $K$ folds $I_1, \ldots, I_K$
2: **for** $k = 1, \ldots, K$ **do**
3:    **Nuisance estimation:**
4:      Estimate $\widehat{e}^{(k)}(\mathbf{X}, \mathbf{z})$ using all observations in $\mathcal{D} \setminus I_k$
5:      Estimate $\widehat{\mathbf{g}}^{(k)}(\mathbf{X}, \mathbf{z})$ using primary observations in $\mathcal{D}\setminus I_k$
6:      Compute out-of-sample predictions $\widehat{e}_i^{(k)}, \widehat{\mathbf{g}}_i^{(k)}$ for each $i \in I_k$
7: **end for**
8: **Target estimation:**
9:    Obtain $\widehat{\boldsymbol{\beta}}$ by minimizing the norm of the average score across folds:

$$\left\|\frac{1}{n}\sum_{k=1}^K\sum_{i \in I_k}\boldsymbol{\psi}_i(\widehat{e}^{(k)}, \widehat{\mathbf{g}}^{(k)}; \widehat{\boldsymbol{\beta}})\right\|$$
$$\leqslant \inf_{\boldsymbol{\beta} \in \mathcal{B}}\left\|\frac{1}{n}\sum_{k=1}^K\sum_{i \in I_k}\boldsymbol{\psi}_i(\widehat{e}^{(k)}, \widehat{\mathbf{g}}^{(k)}; \boldsymbol{\beta})\right\| + o_P(n^{-1/2})$$

10: **Variance estimation:**
11:    Compute $\widehat{\boldsymbol{\psi}}_i = \boldsymbol{\psi}_i(\widehat{e}^{(k)}, \widehat{\mathbf{g}}^{(k)}; \widehat{\boldsymbol{\beta}})$ for each $i \in I_k$, and estimate:

$$\widehat{\boldsymbol{\Sigma}} = \widehat{\mathbf{J}}^{-1}\left(\frac{1}{n}\sum_{k=1}^K\sum_{i \in I_k}\widehat{\boldsymbol{\psi}}_i\widehat{\boldsymbol{\psi}}_i^\top\right)\widehat{\mathbf{J}}^{-1}$$

12:    where $\widehat{\boldsymbol{\psi}}_i = \boldsymbol{\psi}_i(\widehat{e}^{(k)}, \widehat{\mathbf{g}}^{(k)}; \widehat{\boldsymbol{\beta}})$ and $\widehat{\mathbf{J}} = \frac{1}{n}\sum_{k=1}^K\sum_{i \in I_k}\mathbf{X}_i^\top\nabla^2 b(\mathbf{X}_i\widehat{\boldsymbol{\beta}})\mathbf{X}_i$
13: **return** $\widehat{\boldsymbol{\beta}}, \widehat{\boldsymbol{\Sigma}}$

---

**Assumption 5.2** (ML Convergence Rate). There exists $\alpha(n) \downarrow 0$ and $r_1 + r_2 \geqslant 1/2$ such that:

$$\|\widehat{e}(\mathbf{X}, \mathbf{z}) - e^*(\mathbf{X}, \mathbf{z})\|_{P,2} \leqslant \alpha(n)/n^{r_1} \quad (4)$$
$$\|\widehat{\mathbf{g}}(\mathbf{X}, \mathbf{z}) - \mathbf{g}^*(\mathbf{X}, \mathbf{z})\|_{P,2} \leqslant \alpha(n)/n^{r_2} \quad (5)$$

and $\sup_{\mathbf{X}, \mathbf{z}} |\widehat{e}(\mathbf{X}, \mathbf{z}) - e^*(\mathbf{X}, \mathbf{z})| \to_P 0$.

The product rate condition $r_1 + r_2 \geqslant 1/2$ is standard and satisfied by many ML methods including random forests, neural networks, and boosting when the underlying functions have sufficient smoothness.

Under these assumptions, our estimator has the following properties. Define the information matrix $\mathbf{J} := \mathbb{E}[\mathbf{X}^\top\nabla^2 b(\mathbf{X}\boldsymbol{\beta}^*)\mathbf{X}]$.

**Theorem 5.3** (Asymptotic Normality). *Under Assumptions 5.1–5.2,*

$$\sqrt{n}(\widehat{\boldsymbol{\beta}} - \boldsymbol{\beta}^*) \rightsquigarrow N(0, \boldsymbol{\Sigma}^{GAI}) \quad (6)$$

*where* $\boldsymbol{\Sigma}^{GAI} = \mathbf{J}^{-1}\mathbb{E}[\boldsymbol{\psi}\boldsymbol{\psi}^\top]\mathbf{J}^{-1}$.

In typical applications, we assume that $e(\mathbf{X}, \mathbf{z}) = \rho \leqslant 1$ and $w$ is independent of $(\mathbf{X}, \mathbf{y}, \mathbf{z})$. Under this *random labeling* assumption with constant propensity, we prove the following dominance result. We emphasize that this guarantee requires constant propensity; when $e(\mathbf{X}, \mathbf{z})$ varies with covariates, dominance does not hold (see Appendix B for a counterexample). Nevertheless, our experiments with non-constant propensity (Appendix C) show that GAI continues to outperform all benchmarks empirically.

**Corollary 5.4** (Dominance over Primary-Only). *Let $\widehat{\boldsymbol{\beta}}^{\mathrm{P}}$ be estimator obtained through using only human data only, i.e., a solution to $\sum_{i \in \mathcal{D}^{\mathrm{P}}} \nabla_{\boldsymbol{\beta}} \ell(\mathbf{X}, \mathbf{y}; \boldsymbol{\beta}) = 0$. If $e(\mathbf{X}, \mathbf{z}) = \rho$ and $w$ is independent of $(\mathbf{X}, \mathbf{y}, \mathbf{z})$. It holds that $\sqrt{n} \left( \widehat{\boldsymbol{\beta}}^{\mathrm{P}} - \boldsymbol{\beta}^* \right) \rightsquigarrow N \left( \mathbf{0}, \boldsymbol{\Sigma}^{\mathrm{P}} \right)$, where $\boldsymbol{\Sigma}^{\mathrm{P}} = \frac{1}{\rho} \mathbf{J}^{-1} \mathrm{E} \left[ \nabla_{\boldsymbol{\beta}} \ell(\mathbf{X}, \mathbf{y}; \boldsymbol{\beta}^*) \nabla_{\boldsymbol{\beta}} \ell(\mathbf{X}, \mathbf{y}; \boldsymbol{\beta}^*)^{\top} \right] \mathbf{J}^{-1}$ and $\boldsymbol{\Sigma}^{\mathrm{P}} \succeq \boldsymbol{\Sigma}^{\mathrm{GAI}}$. Also, $\boldsymbol{\Sigma}^{\mathrm{P}} \succ \boldsymbol{\Sigma}^{\mathrm{GAI}}$ as long as $\rho < 1$ and*

$$\mathbb{E}\left[ \mathbf{X}^{\top} \left( \nabla b(\mathbf{X}\boldsymbol{\beta}^*) - \mathbb{E}[\mathbf{y} \,|\, \mathbf{X}, \mathbf{z}] \right) \cdot \right.$$
$$\left. \left( \nabla b(\mathbf{X}\boldsymbol{\beta}^*) - \mathbb{E}[\mathbf{y} \,|\, \mathbf{X}, \mathbf{z}] \right)^{\top} \mathbf{X} \right] \succ 0. \quad (7)$$

The proof (Appendix A.2) decomposes the variance into two terms: one depending on $\boldsymbol{\zeta} = \mathbf{X}^{\top}(\nabla b - \mathbf{g}^*)$ (model-prediction error) and one on $\boldsymbol{\pi} = \mathbf{X}^{\top}(\mathbf{g}^* - \mathbf{y})$ (irreducible noise). GAI reduces the first term by factor $\rho$ through auxiliary data, while the second remains unchanged.

It is well-known that estimators based on Neyman-orthogonal score functions are semi-parametric efficient in many settings, for example, when the loss function model is correctly specified as the log-likelihood (Van der Vaart, 2000; Chernozhukov et al., 2018). However, this is not necessarily true in our setting, so the dominance result given above is notable.

# 6. Experiments

## 6.1. Vaccine Conjoint Analysis

Conjoint analysis is a standard tool in marketing for recovering consumer preference parameters from discrete choice data (Green & Srinivasan, 1990; Louviere et al., 2000). Respondents repeatedly choose among product alternatives described by attribute bundles, and researchers infer the underlying part-worth utilities governing these choices.

**Data.** Following Wang et al. (2024), we utilize a vaccine conjoint experiment dataset with 1,971 respondents from Kreps et al. (2020). Each task presents two hypothetical vaccines with 11 attributes and lets respondents select their preferred option. Each respondent was asked to take five tasks in the survey. We estimate the benchmark "ground truth" parameters $\boldsymbol{\beta}^* \in \mathbb{R}^{11}$ by fitting the same logistic

choice model to the full human dataset.[1] We exclude respondents who never select any vaccine option, as many public LLMs are restricted from choosing an opt-out option when generating auxiliary labels; this exclusion is applied consistently both when constructing $\boldsymbol{\beta}^*$ and when forming experimental subsamples.

**AI Representations.** For each choice task, Wang et al. (2024) prompts large language models to generate auxiliary labels and record their output $\mathbf{z} \in \{-1, 0, 1\}$, corresponding to abstaining, choosing option 1, or choosing option 2. We evaluate multiple auxiliary generators that vary both the underlying model and the prompting strategy. Across these configurations, we obtain consistent, qualitatively similar results; for brevity, we report GPT-4o with chain-of-thoughts (CoT) reasoning in the main text and defer the full set of robustness results to Appendix E.1. Excluding abstentions, GPT-4o achieves 54% accuracy. This relatively low accuracy reflects substantial heterogeneity in human preferences: individuals weigh attributes differently based on unobserved constraints and tastes, making aggregate prediction difficult even for strong language models. Importantly, this regime—low accuracy representations—is precisely where treating AI as a surrogate fails, and where GAI's feature-based formulation is valuable.

**Experimental design.** We construct repeated experiments by drawing a primary sample of size $n_P \in \{50, 100, 150, 200\}$ and an auxiliary sample of size $n_A = 1000$ from the available task pool.[2] For each $(n_P, n_A)$ configuration, we run 50 independent trials. Within each trial, the underlying random draw of observations is held fixed across methods to ensure fair comparison. All reported statistics are averaged over the 50 trials.

We compare ten approaches that use the same downstream logistic choice model but differ in how auxiliary representations enter estimation and inference:

1. **Primary**: Maximum likelihood estimation using only human-labeled data.
2. **Naive**: Maximum likelihood estimation pooling human and AI labels without correction.
3. **PPI**: The PPI approach (Angelopoulos et al., 2023a) providing confidence intervals (CIs) that adapts to the prediction quality of AI labels.
4. **PPI++**: A tuning-based, computationally efficient variant of PPI (Angelopoulos et al., 2023b).
5. **RePPI**: Ji et al. (2025) improves the efficiency of PPI using an optimal weight matrix and cross-fitting.
6. **PSPA**: Miao et al. (2025) also improve beyond PPI by

---

[1] All experiments use real data; no semi-synthetic generation is involved.

[2] We randomly sample $n_P/5$ respondents and $n_A/5$ respondents and use all their five choices to compose the primary set and auxiliary set.

utilizing more information from better external AI predictions.

7. **GMM**: Byun et al. (2025) incorporate AI predictions as proxy outcomes via two-step efficient GMM.[3] For methods that use the AI prediction as a surrogate (Naive, PPI, PPI++, RePPI, PSPA, GMM), we use AI predicted labels as $\mathbf{z}$.

8. **Classic-M**: A classical semi-supervised M-estimator (Song et al., 2024) that reweights labeled observations using projection-based optimal weights.

9. **GAI**: Our proposed method uses 5-fold cross-fitting and a regularized logistic regression model to estimate the nuisance function $\mathbf{g}(\mathbf{X}, \mathbf{z})$. In each fold, we fit $\mathbf{g}(\mathbf{X}, \mathbf{z})$ on four-fifths of the primary data using an $\ell_2$-regularized logistic regression (regularization strength $C = 0.05$, maximum 2,000 iterations), and generate out-of-sample predictions for the held-out primary fold as well as for all auxiliary observations. This procedure is repeated across the five folds, and the resulting predictions for each auxiliary observation $\widehat{\mathbf{g}}(\mathbf{X}, \mathbf{z})$ are averaged. Because auxiliary data are never used in training, all predictions for auxiliary observations are strictly out of sample. Predictions for primary observations are also out of sample by construction due to cross-fitting. Since primary and auxiliary observations are randomly sampled from the same underlying population, we set $\widehat{e}(\mathbf{X}, \mathbf{z})$ to the constant $\widehat{e} = n_P/(n_P + n_A)$. Under the MNL model, we then estimate $\widehat{\boldsymbol{\beta}}$ by solving a single optimization problem that minimizes the GAI-adjusted cross-entropy loss:

$$\widehat{\boldsymbol{\beta}} = \arg \min_{\boldsymbol{\beta}} \frac{1}{n} \sum_{i=1}^{n} \left[ -\sum_{j=1}^{J} \widehat{\tau}_{ij} \log p_{ij}(\boldsymbol{\beta}) \right],$$

where

$$\widehat{\tau}_{ij} = \widehat{\mathbf{g}}(X_i, z_i)\left(1 - \frac{w_i}{\widehat{e}}\right) + \frac{w_i y_{ij}}{\widehat{e}},[4]$$

and

$$p_{ij}(\boldsymbol{\beta}) = \frac{\exp(X_{ij}^{\top} \boldsymbol{\beta})}{\sum_{\ell=1}^{J} \exp(X_{i\ell}^{\top} \boldsymbol{\beta})}.$$

This estimator coincides with the general procedure described in Algorithm 1. Finally, we compute standard errors according to Theorem 5.3.

10. **GAI (Emb)**: A variant of GAI where the auxiliary information $\mathbf{z}$ consists of high-dimensional text embeddings of CoT reasoning from GPT-4o, rather than discrete labels. This variant uses $\ell_2$-regularized logistic regression with $C = 0.01$ to learn $\mathbf{g}(\mathbf{X}, \mathbf{z})$.

---

[3]Another variant in Byun et al. (2025) requires generating additional synthetic data, which is distinct from our setting. Therefore, we do not include it as a benchmark.

[4]We clip and normalize $\widehat{\tau}_i$ to derive well-defined probability for numerical stability.

**Main Results.** We evaluate each estimator along three dimensions: (i) point-estimation accuracy relative to $\boldsymbol{\beta}^*$, (ii) empirical coverage of 95% confidence intervals, and (iii) average width of 95% confidence intervals. Point accuracy is summarized by mean absolute percentage error (MAPE):

$$\text{MAPE} = \frac{100}{d} \sum_{j=1}^{d} \frac{|\widehat{\beta}_j - \beta_j^*|}{|\beta_j^*|}. \tag{8}$$

where $d = 11$. Following prior work (Wang et al., 2024), we add a small constant to the denominator since $|\beta_j^*|$ is close to zero to avoid unstable ratios; the constant is set to 1 in the conjoint analysis .

Table 1 reports MAPEs, coverage probabilities, and confidence interval widths across sample sizes. In Panel (a), both GAI and GAI (Emb) achieve the lowest MAPE across all $n_P$, with average error around 16–17%, while the Primary estimator ranges from 19–32% as $n_P$ varies from 200 down to 50. All other benchmarks—including RePPI, PSPA, GMM, and Classic-M—converge toward Primary-level accuracy only at larger $n_P$ and never match GAI. PPI-based methods and RePPI occasionally encounter numerical instability when the primary sample is small, which we denote with "–". Paired $t$-tests confirm that the MAPE reductions achieved by GAI relative to all benchmark methods are statistically significant across most configurations (Appendix E.3).

Panel (b) reports empirical coverage of 95% confidence intervals. GAI and GAI (Emb) exhibit high coverage in all configurations (about 91–99%), while Naive severely undercovers. RePPI substantially undercovers at $n_P = 150$ and 200, and most other benchmarks (PPI++, PSPA, GMM, Classic-M) fall below nominal coverage. Panel (c) reports average confidence interval width. GAI maintains the tightest intervals among methods that achieve valid coverage, while PPI intervals are markedly wider in small-$n_P$ settings. GAI (Emb) uses richer representations and achieves slightly higher coverage at the cost of moderately wider intervals. Ablation studies examining the sources of GAI's improvement are provided in Appendix D.

Figure 2 provides a complementary distributional view for $n_P = 200$. Panel (a) plots the distribution of trial-level confidence interval coverage averaged across parameters. Both GAI and PPI exhibit coverage tightly concentrated near the nominal level, while PPI++ shows slight undercoverage. Panel (b) plots the distribution of confidence interval widths and illustrates the wider dispersion of intervals for PPI-based estimators. Taken together, GAI produces narrower confidence intervals without sacrificing coverage validity. Additional robustness checks that vary the auxiliary generator (model and prompting) are reported in Appendix E.1.

*Table 1.* Benchmark Comparison for Conjoint Analysis: MAPEs, coverage probabilities, and CI widths

| Method | $n_P$=50 | $n_P$=100 | $n_P$=150 | $n_P$=200 |
|---|---|---|---|---|
| **Panel (a): MAPE (%)** | | | | |
| Primary | 32.02 | 25.27 | 19.67 | 19.01 |
| Naive | 48.52 | 45.75 | 43.33 | 40.29 |
| PPI | – | 44.57 | 32.96 | 29.21 |
| PPI++ | – | 29.69 | 22.96 | 21.00 |
| RePPI | – | – | 41.62 | 25.73 |
| PSPA | 49.86 | 29.92 | 22.69 | 20.79 |
| GMM | 57.80 | 31.70 | 22.95 | 21.00 |
| Classic-M | 79.32 | 33.25 | 23.81 | 21.45 |
| **GAI** | **16.86** | **17.52** | **15.97** | **16.64** |
| **GAI (Emb)** | **16.50** | **17.23** | **15.73** | **16.24** |
| **Panel (b): 95% CI Coverage Probability (%)** | | | | |
| Primary | 97.82 | 92.73 | 92.55 | 88.00 |
| Naive | 26.55 | 26.36 | 26.36 | 27.09 |
| PPI | 99.82 | 98.00 | 95.82 | 92.91 |
| PPI++ | 94.55 | 89.09 | 90.91 | 85.09 |
| RePPI | – | 82.36 | 74.73 | 77.82 |
| PSPA | 93.82 | 90.00 | 91.45 | 85.27 |
| GMM | 88.91 | 86.73 | 89.64 | 84.73 |
| Classic-M | 87.82 | 90.00 | 90.91 | 86.91 |
| **GAI** | **99.82** | **97.09** | **94.73** | **90.91** |
| **GAI (Emb)** | **99.45** | **98.55** | **96.91** | **94.55** |
| **Panel (c): 95% CI Width** | | | | |
| Primary | 2.43 | 1.51 | 1.19 | 1.01 |
| Naive | 0.55 | 0.53 | 0.51 | 0.49 |
| PPI | 9.28 | 2.71 | 1.94 | 1.58 |
| PPI++ | 2.44 | 1.52 | 1.19 | 1.00 |
| RePPI | – | – | 1.46 | 1.01 |
| PSPA | 2.47 | 1.53 | 1.20 | 1.01 |
| GMM | – | 1.49 | 1.17 | 1.00 |
| Classic-M | 3.57 | 1.74 | 1.29 | 1.07 |
| **GAI** | **1.94** | **1.36** | **1.11** | **0.96** |
| **GAI (Emb)** | **2.11** | **1.50** | **1.21** | **1.05** |

**Notes**: A "–" symbol indicates cases where the value exceeds 1,000 or the method encounters numerical instability in small primary samples.

## 6.2. Health Insurance Census Analysis

We next consider a setting where auxiliary representations are continuous and well calibrated. This experiment follows the private health insurance coverage example in Angelopoulos et al. (2023a) and complements the conjoint analysis by shifting to a high-quality prediction regime.

**Data.** We use the California census healthcare dataset with 318,215 individuals to investigate the relationship between private health insurance coverage ($y \in \{0, 1\}$) and income using a logistic model. Ground truth of logistic regression coefficients $\boldsymbol{\beta}^* \in \mathbb{R}^2$ is estimated using all data.

**AI Representations.** Auxiliary representations are produced by Angelopoulos et al. (2023a) using a gradient boosting classifier trained on a richer feature set, including in-

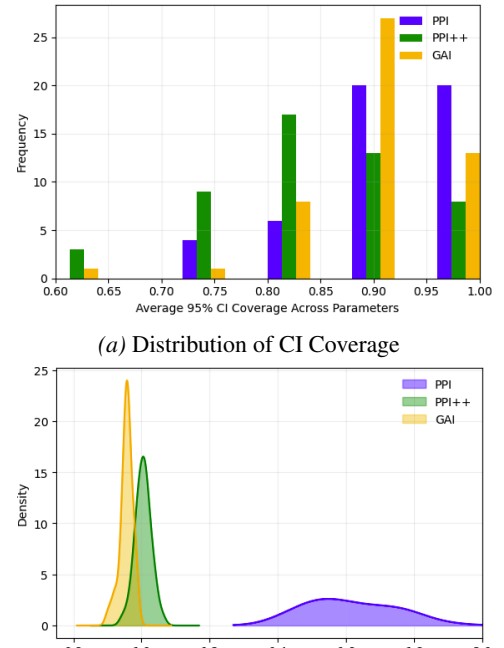

*(a)* Distribution of CI Coverage

*(b)* Distribution of CI Width

*Figure 2.* Comparison between PPI, PPI++ and GAI for Conjoint Analysis ($n_P = 200$)

come, race, sex, and other covariates. The model outputs predicted probabilities $\mathbf{z} = P(y = 1 \,|\, \mathbf{X})$ and achieves approximately 85% accuracy. Unlike the AI-generated data in the conjoint analysis in Section 6.1, this representation is well-calibrated.

**GAI implementation.** Because $\mathbf{z}$ already represents an accurate estimate of the conditional mean based on richer features, learning an additional $\mathbf{g}(\mathbf{X}, \mathbf{z})$ offers little benefit. Therefore, we set $\mathbf{g}(\mathbf{X}, \mathbf{z}) = \mathbf{z}$ in this experiment. Similar to the conjoint analysis, $e(\mathbf{X}, \mathbf{z})$ is set at as a constant $n_P/(n_P + n_A)$. This yields a simplified version of our estimator that does not require first-stage estimation of nuisance parameters (i.e., $\mathbf{g}(\mathbf{X}, \mathbf{z})$ and $e(\mathbf{X}, \mathbf{z})$) or cross-fitting.

**Experimental Design.** We vary the primary sample sizes $n_P \in \{100, 250, 500, 750, 1000\}$, with a fixed auxiliary sample size $n_A = 2000$. For each configuration, we run 50 independent trials. In each trial, the same random subsample is reused across all methods to isolate estimator differences. We compare the same set of benchmarks as in the conjoint analysis. We compute MAPE, empirical 95% CI coverage, and average 95% CI width as in the conjoint analysis.

**Main Results.** Table 2 summarizes estimation accuracy and inference performance. Panel (a) shows that GAI substantially reduces MAPE relative to all benchmarks across all $n_P$, with errors around 140–160%, while the Primary estimator remains above 290% even at $n_P = 1000$. The

*Table 2.* Benchmark Comparison for Census Analysis: MAPEs, coverage probabilities, and CI widths

| Method | $n_P$=100 | $n_P$=250 | $n_P$=500 | $n_P$=750 | $n_P$=1000 |
|---|---|---|---|---|---|
| **MAPE (%)** | | | | | |
| Primary | 979.83 | 563.85 | 480.99 | 320.68 | 291.80 |
| Naive | 407.95 | 384.70 | 324.56 | 263.13 | 240.38 |
| PPI | 821.83 | 518.82 | 383.92 | 295.40 | 270.66 |
| PPI++ | 866.27 | 508.75 | 404.75 | 299.52 | 282.43 |
| RePPI | – | 589.38 | 433.31 | 301.30 | 273.30 |
| PSPA | 804.65 | 484.00 | 379.09 | 291.29 | 274.20 |
| GMM | 809.43 | 480.11 | 357.46 | 285.53 | 270.02 |
| Classic-M | – | 524.55 | 410.19 | 301.98 | 265.74 |
| **GAI** | **161.37** | **147.93** | **147.38** | **139.65** | **141.49** |
| **95% CI Coverage Probability (%)** | | | | | |
| Primary | 80.00 | 78.00 | 78.00 | 82.00 | 78.00 |
| Naive | 30.00 | 31.00 | 34.00 | 38.00 | 38.00 |
| PPI | 88.00 | 94.00 | 95.00 | 96.00 | 94.00 |
| PPI++ | 83.00 | 88.00 | 91.00 | 93.00 | 89.00 |
| RePPI | 78.00 | 89.00 | 88.00 | 92.00 | 91.00 |
| PSPA | 83.00 | 87.00 | 91.00 | 91.00 | 89.00 |
| GMM | 83.00 | 85.00 | 91.00 | 92.00 | 90.00 |
| Classic-M | 89.00 | 90.00 | 92.00 | 93.00 | 93.00 |
| **GAI** | **100.00** | **100.00** | **100.00** | **99.00** | **99.00** |
| **95% CI width ($10^{-5}$)** | | | | | |
| Primary | 2.16 | 1.31 | 0.92 | 0.74 | 0.65 |
| Naive | 0.61 | 0.58 | 0.53 | 0.49 | 0.46 |
| PPI | 2.48 | 1.85 | 1.29 | 1.10 | 0.96 |
| PPI++ | 2.24 | 1.70 | 1.24 | 1.02 | 0.89 |
| RePPI | 4.90 | 1.90 | 1.30 | 1.10 | 0.90 |
| PSPA | 1.90 | 1.50 | 1.20 | 1.00 | 0.90 |
| GMM | 1.90 | 1.50 | 1.20 | 1.00 | 0.90 |
| Classic-M | – | 1.80 | 1.30 | 1.10 | 1.00 |
| **GAI** | **2.64** | **1.79** | **1.28** | **1.08** | **0.93** |

**Notes**: A "–" symbol indicates cases where the value exceeds 10,000 or the method encounters numerical instability in small primary samples.

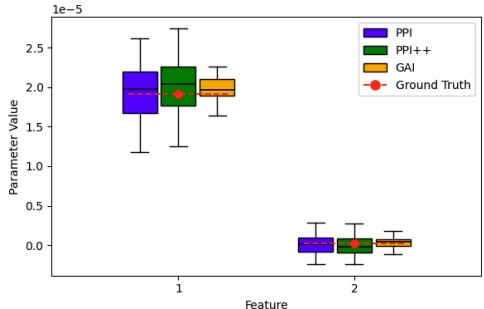

*Figure 3.* Distribution of PPI, PPI++ and GAI Estimates for Census Analysis ($n_P = 1000$)

by $10^{-5}$). GAI confidence intervals are of similar magnitude to PPI, PPI++, and the other benchmarks, indicating that the coverage improvements are not driven by overly conservative intervals. Formal statistical tests show that GAI achieves these gains in coverage and precision without sacrificing inferential validity (Appendix E.3).

Taken together, these two empirical settings span materially different auxiliary-data regimes: discrete and inaccurate LLM outputs in the vaccine conjoint, versus continuous and well-calibrated probabilities in the census application. Additional implementation details and robustness checks are provided in Appendix E.1.

## 7. Conclusion

We proposed a new perspective on how AI representations can be used for statistical inference: as *informative features* rather than *outcome proxies*. Formalized through Neyman orthogonality, this perspective enables valid inference across a wide range of AI representation quality and variety—from inaccurate, discrete LLM responses to well-calibrated probabilistic representations.

Theoretically, we establish asymptotic normality and prove that, under random labeling, our estimator strictly dominates primary-only estimation under random selection whenever AI representations carry incremental information. Unlike PPI, which treats AI predictions as surrogate realizations of the outcome, our GAI framework incorporates AI outputs as features to learn human labels, thereby allowing unrestricted AI representations to contribute information without requiring them to approximate human labels. Empirically, GAI performs strongly across both auxiliary-data regimes and significantly outperforms PPI-based approaches. In a vaccine conjoint analysis with inaccurate LLM labels (54% accuracy), GAI reduces MAPE from 32.02% (primary-only) to 16.86%, while PPI exhibits substantially larger error and wider intervals, and GAI maintains nominal coverage. In a census application with high-quality representations (85% accuracy), GAI achieves sizable accuracy gains and valid

benchmarks reduce error relative to Primary but do not match GAI in these runs. Paired $t$-tests confirm that the MAPE improvements of GAI over all benchmark methods are statistically significant across all primary sample sizes considered (Appendix E.3).

Figure 3 provides an additional distributional perspective for these results. The boxplots show that GAI estimates are more tightly concentrated around the ground truth for both coefficients. In contrast, PPI and PPI++ exhibit wider dispersion.

Table 2 Panel (b) reports coverage probabilities. Primary and Naive exhibit severe undercoverage, consistent with their severely biased point estimates. All other benchmarks—PPI, PPI++, RePPI, PSPA, GMM, and Classic-M—improve coverage relative to Primary but remain below 95% nominal level in most configurations, while GAI attains coverage of 99–100% throughout. Panel (c) reports CI widths (scaled

coverage comparable to or exceeding PPI and PPI++, without relying on inflated confidence intervals.

Our framework allows researchers to combine a small number of expensive human labels with abundant AI-generated data while preserving inference validity, which can be applied to market research, medicine, and social science. In practical terms, achieving the precision of 200 human labels using only 50 human labels supplemented with 1,000 AI representations represents a cost reduction around 75% at typical annotation rates.

## Impact Statement

By enabling statistically valid use of AI-generated data, this work has the potential to substantially reduce annotation costs in market research, medicine, and social science. At the same time, reliance on AI augmentation raises concerns about labor displacement and the propagation of demographic biases embedded in AI systems. While our framework corrects for predictive bias in expectation, it does not guarantee fairness across subpopulations. We therefore strongly recommend conducting subgroup-level audits—stratified by demographic attributes such as race, gender, and age—to verify that AI representations do not introduce or amplify disparities in estimation accuracy. Such audits should be performed before deployment in any high-stakes or equity-sensitive application.

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

## A. Detailed Proofs

### A.1. Proof of Theorem 5.3

In the proof, for simplicity we assume without loss of generality that we can split the dataset into $I$ and $I^c$, each of size $n$. We obtain $\widehat{e}(\cdot)$ and $\widehat{\mathbf{g}}(\cdot)$ on $I^c$ and estimate $\widehat{\boldsymbol{\beta}}$ on $I$. Let us define

$$\boldsymbol{\tau}(\Xi; e, \mathbf{g}) := \mathbf{X}^\top \left[ \mathbf{g}(\mathbf{X}, \mathbf{z}) - \frac{w}{e(\mathbf{X}, \mathbf{z})} (\mathbf{g}(\mathbf{X}, \mathbf{z}) - \mathbf{y}) \right] \tag{9}$$

so $\boldsymbol{\psi}(\Xi; e, \mathbf{g}; \boldsymbol{\beta}) = \mathbf{X} \nabla b(\mathbf{X}\boldsymbol{\beta}) - \boldsymbol{\tau}(\Xi; e, \mathbf{g})$ for all $\boldsymbol{\beta} \in \mathcal{B}$. For simplicity, following the empirical process literature (see Van der Vaart 2000; Vaart & Wellner 2023), we often use the shorthand notation $\mathbb{P}_n f$ to denote the empirical expectation of a function $f$ based on data in $I$, $\mathbb{G}_n f$ to denote the empirical process based on data in $I$, and $Pf$ to denote its population expectation. Also, given the boundedness assumption on $\mathcal{X}$, we assume that there is $C$ such that for any $\mathbf{X} \in \mathcal{X}$, $\|\mathbf{X}\|, \|\mathbf{X}\|_F \leqslant C$. We let $B(\mathbf{x}, \epsilon)$ to denote an open ball of radius $\epsilon$ around the vector $\mathbf{x} \in \mathbb{R}^q$.

The next lemma establishes that our score function given in 3 is valid and states the key consequence of its Neyman orthogonal design.

**Lemma A.1** (**Score Function and Neyman Orthogonality**). *It holds that*

$$P\boldsymbol{\psi}(e^*, \mathbf{g}^*; \boldsymbol{\beta}^*) = 0 \ \text{ and } \ \mathbb{P}_n\boldsymbol{\tau}(\widehat{e}, \widehat{\mathbf{g}}) - \mathbb{P}_n\boldsymbol{\tau}(e^*, \mathbf{g}^*) = o_P(n^{-1/2}).$$

The next two key lemmas explores the concavity of our loss function. The first one clarifies the convergence rate of the empirical score functions. Building on this, we present a consistency proof.

**Lemma A.2** (**Rates of Empirical Scores**). *It holds that*

$$\left\| \mathbb{P}_n\boldsymbol{\psi}\left(e^*, \mathbf{g}^*; \widehat{\boldsymbol{\beta}}\right) \right\|, \ \inf_{\boldsymbol{\beta} \in \mathcal{B}} \left\| \mathbb{P}_n\boldsymbol{\psi}\left(\widehat{e}, \widehat{\mathbf{g}}; \boldsymbol{\beta}\right) \right\| = o_P(1/\sqrt{n}).$$

**Lemma A.3** (**Consistency**). $\widehat{\boldsymbol{\beta}} \to_P \boldsymbol{\beta}^*$.

With these lemmas we finish the proof of the main result. For ease of exposition, from this point on we suppress $e^*$ and $\mathbf{g}^*$ in $\boldsymbol{\psi}(\cdot; e^*, \mathbf{g}^*; \boldsymbol{\beta})$ and write it as $\boldsymbol{\psi}(\cdot; \boldsymbol{\beta})$. We first establish that the score function is locally Lipschitz. Indeed, fix an arbitrary $\epsilon$ and consider $\boldsymbol{\beta}_1, \boldsymbol{\beta}_2 \in B(\boldsymbol{\beta}^*, \epsilon)$, we have

$$\|\boldsymbol{\psi}(\Xi; \boldsymbol{\beta}) - \boldsymbol{\psi}(\Xi; \boldsymbol{\beta}_2)\| = \left\| \mathbf{X}^\top \nabla b(\mathbf{X}\boldsymbol{\beta}_1) - \mathbf{X}^\top \nabla b(\mathbf{X}\boldsymbol{\beta}_2) \right\|$$

$$= \left\| \mathbf{X}^\top \int_0^1 \nabla_{\boldsymbol{\theta}}^2 b\left(\mathbf{X}\left(\boldsymbol{\beta}_1 + t(\boldsymbol{\beta}_2 - \boldsymbol{\beta}_1)\right)\right) dt(\boldsymbol{\beta}_2 - \boldsymbol{\beta}_1) \right\| \leqslant \left\| \mathbf{X}^\top \int_0^1 \nabla_{\boldsymbol{\theta}}^2 b\left(\mathbf{X}\left(\boldsymbol{\beta}_1 + t(\boldsymbol{\beta}_2 - \boldsymbol{\beta}_1)\right)\right) dt \right\| \|\boldsymbol{\beta}_2 - \boldsymbol{\beta}_1\|$$

$$\leqslant \|\mathbf{X}\| \int_0^1 \left\| \nabla_{\boldsymbol{\theta}}^2 b\left(\mathbf{X}\left(\boldsymbol{\beta}_1 + t(\boldsymbol{\beta}_2 - \boldsymbol{\beta}_1)\right)\right) \right\| dt \|\boldsymbol{\beta}_2 - \boldsymbol{\beta}_1\| \leqslant \|\boldsymbol{\beta}_2 - \boldsymbol{\beta}_1\| \left( \|\mathbf{X}\| \sup_{\boldsymbol{\beta} \in \overline{B(\boldsymbol{\beta}^*, \epsilon)}} \|\nabla_{\boldsymbol{\theta}}^2 b(\mathbf{X}\boldsymbol{\beta})\| \right)$$

$$\leqslant C_1 \|\mathbf{X}\| \|\boldsymbol{\beta}_2 - \boldsymbol{\beta}_1\|$$

where we define $C_1 := \sup_{\boldsymbol{\beta} \in \overline{B(\boldsymbol{\beta}^*, \epsilon)}, \mathbf{X} \in \breve{\mathcal{X}}} \|\nabla_{\boldsymbol{\theta}}^2 b(\mathbf{X}\boldsymbol{\beta})\| < \infty$. Indeed, the supreme must be convex by the compactness of $\breve{\mathcal{X}}$ and $\overline{B(\boldsymbol{\beta}^*, \epsilon)}$ and the continuity of $\nabla_{\boldsymbol{\theta}}^2 b(\cdot)$. Clearly, $PC_1^2\|\mathbf{X}\|^2 < \infty$ Therefore, by Example 19.7 and Theorem 19.5 in (Van der Vaart, 2000), $\{\boldsymbol{\psi}(\cdot; \boldsymbol{\beta}) : \boldsymbol{\beta} \in B(\boldsymbol{\beta}^*, \epsilon)\}$ forms a Donsker class. By Lemma A.3, $\widehat{\boldsymbol{\beta}} \to_P \boldsymbol{\beta}^*$. Then, By Theorem 19.9 in in (Van der Vaart, 2000), it holds that $\mathbb{G}_n\boldsymbol{\psi}(\widehat{\boldsymbol{\beta}}) - \mathbb{G}_n\boldsymbol{\psi}(\boldsymbol{\beta}^*) \to_P 0$. Thus,

$$\sqrt{n}(P\boldsymbol{\psi}(\widehat{\boldsymbol{\beta}}) - P\boldsymbol{\psi}(\boldsymbol{\beta}^*)) = \mathbb{G}_n\boldsymbol{\psi}(\widehat{\boldsymbol{\beta}}) - \mathbb{G}_n\boldsymbol{\psi}(\boldsymbol{\beta}^*) - \sqrt{n}\mathbb{P}_n\boldsymbol{\psi}\left(\widehat{\boldsymbol{\beta}}\right) + \sqrt{n}\mathbb{P}_n\boldsymbol{\psi}\left(\boldsymbol{\beta}^*\right) \overset{(a)}{=} \sqrt{n}\mathbb{P}_n\boldsymbol{\psi}\left(\boldsymbol{\beta}^*\right) + o_P(1),$$

where (a) follows from Lemma A.2. We apply the Taylor expansion on the left-side of the equation, by continuous mapping, we have

$$\sqrt{n} \left[ \left( P\mathbf{X}^\top \nabla b_{\boldsymbol{\theta}}(\mathbf{X}\boldsymbol{\beta}^*)\mathbf{X} \right)(\widehat{\boldsymbol{\beta}} - \boldsymbol{\beta}^*) + o_P(1) \left\| \widehat{\boldsymbol{\beta}} - \boldsymbol{\beta}^* \right\| \right] = \sqrt{n}\mathbb{P}_n\boldsymbol{\psi}\left(\boldsymbol{\beta}^*\right) + o_P(1) = O_P(1). \tag{10}$$

As shown in the proof of Lemma A.2, $P\mathbf{X}^\top \nabla b_{\boldsymbol{\theta}}(\mathbf{X}\boldsymbol{\beta}^*)\mathbf{X}$ is invertible. Therefore, (10) implies that $\sqrt{n}\left\|\widehat{\boldsymbol{\beta}} - \boldsymbol{\beta}^*\right\| = O_P(1)$. Therefore, applying (10) again,

$$\sqrt{n}\left(\widehat{\boldsymbol{\beta}} - \boldsymbol{\beta}^*\right) = \left(P\mathbf{X}^\top \nabla b_{\boldsymbol{\theta}}(\mathbf{X}\boldsymbol{\beta}^*)\mathbf{X}\right)^{-1}\sqrt{n}\mathbb{P}_n\boldsymbol{\psi}(\boldsymbol{\beta}^*) + o_P(1),$$

as desired.

## A.2. Proof of Corollary 5.4

To start, we first note that the empirical sums of score functions, i.e., $\sum_{i\in\mathcal{D}^P}\nabla_{\boldsymbol{\beta}}\ell(\mathbf{X},\mathbf{y};\boldsymbol{\beta}) = 0$, is equivalent to $\sum_{i=1}^n w_i\nabla_{\boldsymbol{\beta}}\ell(\mathbf{X},\mathbf{y};\boldsymbol{\beta}^*) = 0$. Note that by independence of $w$ with respect to other random variables, it holds that

$$\mathrm{E}\left[w\nabla_{\boldsymbol{\beta}}\ell(\mathbf{X},\mathbf{y};\boldsymbol{\beta})\right] = \rho\mathrm{E}\left[\nabla_{\boldsymbol{\beta}}\ell(\mathbf{X},\mathbf{y};\boldsymbol{\beta}^*)\right] = 0.$$

Thus, under the preset assumptions, it is easy to show that with probability approaching one, the solution to this equation, i.e., $\widehat{\boldsymbol{\beta}}^P$, exists, and $\sqrt{n}\left(\widehat{\boldsymbol{\beta}} - \boldsymbol{\beta}^*\right) \rightsquigarrow N\left(\mathbf{0}, \boldsymbol{\Sigma}^P\right)$. We skip the proof. We further note that $\nabla_{\boldsymbol{\beta}}\ell(\mathbf{X},\mathbf{y};\boldsymbol{\beta}^*) := \boldsymbol{\zeta}(\mathbf{X},\mathbf{z}) + \boldsymbol{\pi}(\mathbf{X},\mathbf{y},\mathbf{z})$ where $\boldsymbol{\zeta}(\mathbf{X},\mathbf{z}) := \mathbf{X}^\top\left(\nabla_{\boldsymbol{\theta}}b(\mathbf{X}\boldsymbol{\beta}^*) - \mathbf{g}(\mathbf{X},\mathbf{z})\right)$ and $\boldsymbol{\pi}(\mathbf{X},\mathbf{y},\mathbf{z}) = \mathbf{X}^\top\left(\mathbf{g}(\mathbf{X},\mathbf{z}) - \mathbf{y}\right)$ and, because $\mathrm{E}\left[\mathbf{g}(\mathbf{X},\mathbf{z}) - \mathbf{y}|\mathbf{X},\mathbf{z}\right] = 0$, we have

$$\boldsymbol{\Sigma}^P = \frac{1}{\rho}\left(\mathbf{J}^{-1}\mathrm{E}\left[\boldsymbol{\zeta}(\mathbf{X},\mathbf{z})\boldsymbol{\zeta}(\mathbf{X},\mathbf{z})^\top\right]\mathbf{J}^{-1} + \mathbf{J}^{-1}\mathrm{E}\left[\boldsymbol{\pi}(\mathbf{X},\mathbf{y},\mathbf{z})\boldsymbol{\pi}(\mathbf{X},\mathbf{y},\mathbf{z})^\top\right]\mathbf{J}^{-1}\right). \tag{11}$$

Furthermore, observe that $\boldsymbol{\psi}(\Xi;e,\mathbf{g};\boldsymbol{\beta}) = \boldsymbol{\zeta}(\mathbf{X},\mathbf{z}) + \frac{w}{\rho}\boldsymbol{\pi}(\mathbf{X},\mathbf{y},\mathbf{z})$, so

$$\boldsymbol{\Sigma}^{\mathrm{GAI}} = \mathbf{J}^{-1}\mathrm{E}\left[\boldsymbol{\zeta}(\mathbf{X},\mathbf{z})\boldsymbol{\zeta}(\mathbf{X},\mathbf{z})^\top\right]\mathbf{J}^{-1} + \frac{1}{\rho}\mathbf{J}^{-1}\mathrm{E}\left[\boldsymbol{\pi}(\mathbf{X},\mathbf{y},\mathbf{z})\boldsymbol{\pi}(\mathbf{X},\mathbf{y},\mathbf{z})^\top\right]\mathbf{J}^{-1}. \tag{12}$$

The conclusion follows straightforwardly given (11) and (12).

## A.3. Proof of Lemma A.1

To begin, first we note that by the definition of $\boldsymbol{\tau}(e^*,\mathbf{g}^*)$

$$P\boldsymbol{\tau}(e^*,\mathbf{g}^*) = \mathrm{E}\left[\mathbf{X}^\top\left[\mathbf{g}^*(\mathbf{X},\mathbf{z}) - \frac{w}{e^*(\mathbf{X},\mathbf{z})}(\mathbf{g}^*(\mathbf{X},\mathbf{z}) - \mathbf{y})\right]\right]$$

$$\stackrel{(a)}{=} \mathrm{E}\left[\mathbf{X}^\top\left[\mathbf{g}^*(\mathbf{X},\mathbf{z}) - \frac{\mathrm{E}[w|\mathbf{X},\mathbf{z}]}{e^*(\mathbf{X},\mathbf{z})}(\mathbf{g}^*(\mathbf{X},\mathbf{z}) - \mathrm{E}[\mathbf{y}|\mathbf{X},\mathbf{z}])\right]\right] \stackrel{(b)}{=} \mathrm{E}\left[\mathbf{X}^\top\mathrm{E}[\mathbf{y}|\mathbf{X},\mathbf{z}]\right] = \mathrm{E}\left[\mathbf{X}^\top\mathbf{y}\right],$$

where (a) follows from the law of iterated expectations and the conditional independence of $w$ and $\mathbf{y}$, and (b) follows from the definition that $\mathrm{E}[w|\mathbf{X},\mathbf{z}] = e^*(\mathbf{X},\mathbf{z})$. Therefore,

$$P\boldsymbol{\psi}(e^*,\mathbf{g}^*;\boldsymbol{\beta}^*) = \mathrm{E}\left[\mathbf{X}^\top b(\mathbf{X}\boldsymbol{\beta}) - \boldsymbol{\tau}(\Xi;e^*,\mathbf{g}^*)\right] = \mathrm{E}\left[\mathbf{X}^\top b(\mathbf{X}\boldsymbol{\beta}) - \mathbf{X}^\top\mathbf{y}\right] = 0.$$

This proves the first part. For the second, we notice that

$$\sqrt{n}\left(\mathbb{P}_n\boldsymbol{\tau}(\widehat{e},\widehat{\mathbf{g}}) - \mathbb{P}_n\boldsymbol{\tau}(e^*,\mathbf{g}^*)\right)$$

$$= \underbrace{\sqrt{n}\mathbb{P}_n\left[\left(1 - \frac{w}{e^*}\right)\mathbf{X}^\top(\widehat{\mathbf{g}} - \mathbf{g}^*)\right]}_{(\mathrm{I})} + \underbrace{\sqrt{n}\mathbb{P}_n\left[\frac{w(\widehat{e} - e^*)}{\widehat{e}e^*}\mathbf{X}^\top(\mathbf{g}^* - \mathbf{y})\right]}_{(\mathrm{II})} + \underbrace{\sqrt{n}\mathbb{P}_n\left[\frac{w(\widehat{e} - e^*)}{\widehat{e}e^*}\mathbf{X}^\top(\widehat{\mathbf{g}} - \mathbf{g}^*)\right]}_{(\mathrm{III})}.$$

We treat each term separately. For (I), we note that

$$\mathrm{E}\left[\left(1 - \frac{w}{e^*(\mathbf{X},\mathbf{z})}\right)\mathbf{X}^\top(\widehat{\mathbf{g}}(\mathbf{X},\mathbf{z}) - \mathbf{g}^*(\mathbf{X},\mathbf{z}))\Big|I^c\right]$$

$$= \mathrm{E}\left[\left(1 - \frac{\mathrm{E}[w|\mathbf{X},Z,I^c]}{e^*(\mathbf{X},\mathbf{z})}\right)\mathbf{X}^\top(\widehat{\mathbf{g}}(\mathbf{X},\mathbf{z}) - \mathbf{g}^*(\mathbf{X},\mathbf{z}))\Big|I^c\right] = 0 \tag{13}$$

so (I) has zero mean. Therefore, by Markov inequality

$$
\begin{aligned}
\mathrm{P}\left(\|(\mathrm{I})\| \geqslant \epsilon \mid I^c\right) &\leqslant \frac{\mathrm{E}[\|(\mathrm{I})\|^2 \mid I^c]}{\epsilon^2} = \sum_{j=1}^{d} \mathrm{Var}\left[\sqrt{n}\mathbb{P}_n\left[\left(1 - \frac{w}{e^*}\right)\mathbf{x}_{(j)}^\top(\widehat{\mathbf{g}} - \mathbf{g}^*)\right]\,\Big|\,I^c\right]\Big/\epsilon^2 \\
&= \sum_{j=1}^{d} \mathrm{Var}\left[\left(1 - \frac{w}{e^*(\mathbf{X},\mathbf{z})}\right)\mathbf{x}_{(j)}^\top(\widehat{\mathbf{g}}(\mathbf{X},\mathbf{z}) - \mathbf{g}^*(\mathbf{X},\mathbf{z}))\,\Big|\,I^c\right]\Big/\epsilon^2 \\
&= \mathrm{E}\left[\left\|\left(1 - \frac{w}{e^*(\mathbf{X},\mathbf{z})}\right)\mathbf{X}^\top(\widehat{\mathbf{g}}(\mathbf{X},\mathbf{z}) - \mathbf{g}^*(\mathbf{X},\mathbf{z}))\right\|^2\,\Big|\,I^c\right]\Big/\epsilon^2. \qquad (14)
\end{aligned}
$$

From here, we note that

$$
\begin{aligned}
\left\|\left(1 - \frac{w}{e^*(\mathbf{X},\mathbf{z})}\right)\mathbf{X}^\top(\widehat{\mathbf{g}}(\mathbf{X},\mathbf{z}) - \mathbf{g}^*(\mathbf{X},\mathbf{z}))\right\|_{P,2}^2 &\leqslant \frac{4}{\kappa^2}\left\|\mathbf{X}^\top(\widehat{\mathbf{g}}(\mathbf{X},\mathbf{z}) - \mathbf{g}^*(\mathbf{X},\mathbf{z}))\right\|_{P,2}^2 \\
&\leqslant \frac{4C^2}{\kappa^2}\left\|\widehat{\mathbf{g}}(\mathbf{X},\mathbf{z}) - \mathbf{g}^*(\mathbf{X},\mathbf{z})\right\|_{P,2}^2 \overset{(c)}{\to} 0,
\end{aligned}
$$

which implies the RHS of (14) converges to zero in probability by the Markov's inequality. Here (c) follows from Assumption 5.2. Since $\mathrm{P}\left(\|(\mathrm{I})\| \geqslant \epsilon \mid I^c\right)$ is bounded therefore uniformly integrable, we can take expectation and conclude that (I) is $o_P(1)$. An argument similar to (13) shows that (II) has zero mean. An argument similar to (14) shows that

$$
\begin{aligned}
\mathrm{P}\left(\|(\mathrm{II})\| \geqslant \epsilon \mid I^c\right) &\leqslant \mathrm{E}\left[\left\|\frac{w(\widehat{e}(\mathbf{X},\mathbf{z}) - e^*(\mathbf{X},\mathbf{z}))}{\widehat{e}(\mathbf{X},\mathbf{z})e^*(\mathbf{X},\mathbf{z})}\mathbf{X}^\top(\mathbf{g}^*(\mathbf{X},\mathbf{z}) - \mathbf{y})\right\|^2\,\Big|\,I^c\right]\Big/\epsilon^2 \\
&\overset{(d)}{=} \frac{2}{\kappa^2\epsilon^2}\mathrm{E}\left[\left\|w(\widehat{e}(\mathbf{X},\mathbf{z}) - e^*(\mathbf{X},\mathbf{z}))\mathbf{X}^\top(\mathbf{g}^*(\mathbf{X},\mathbf{z}) - \mathbf{y})\right\|^2\,\Big|\,I^c\right] + o_P(1) \\
&= \frac{2}{\kappa^2\epsilon^2}\mathrm{E}\left[|\widehat{e}(\mathbf{X},\mathbf{z}) - e^*(\mathbf{X},\mathbf{z})|^2\sum_{j=1}^{d}\mathbf{x}_{(j)}^\top\mathrm{Cov}(\mathbf{y}|\mathbf{X},\mathbf{z},I^c)\mathbf{x}_{(j)}\,\Big|\,I^c\right] + o_P(1) \\
&\overset{(e)}{\leqslant} \frac{2\tilde{\sigma}^2}{\kappa^2\epsilon^2}\mathrm{E}\left[|\widehat{e}(\mathbf{X},\mathbf{z}) - e^*(\mathbf{X},\mathbf{z})|^2\sum_{j=1}^{d}\mathbf{x}_{(j)}^\top\mathbf{x}_{(j)}\,\Big|\,I^c\right] + o_P(1) \\
&= \frac{2\tilde{\sigma}^2}{\kappa^2\epsilon^2}\mathrm{E}\left[|\widehat{e}(\mathbf{X},\mathbf{z}) - e^*(\mathbf{X},\mathbf{z})|^2\|\mathbf{X}\|_F^2\,\Big|\,I^c\right] + o_P(1) \\
&= \frac{2C^2\tilde{\sigma}^2}{\kappa^2\epsilon^2}\mathrm{E}\left[|\widehat{e}(\mathbf{X},\mathbf{z}) - e^*(\mathbf{X},\mathbf{z})|^2\,\Big|\,I^c\right] + o_P(1), \qquad (15)
\end{aligned}
$$

where (d) follows because for all $\mathbf{X}$ and $\mathbf{z}$, (1) by assumption $e^*(\mathbf{X},\mathbf{z}) \geqslant \kappa$ and (2) Assumption 5.2 states that $\sup_{\mathbf{X},\mathbf{z}}|\widehat{e}(\mathbf{X},\mathbf{z}) - e^*(\mathbf{X},\mathbf{z})| \to_P 0$, so with probability approaching one it holds that $\widehat{e}(\mathbf{X},\mathbf{z}) \geqslant \kappa/2$. (e) holds because the distribution of $\mathbf{y}$ is independent of $I^c$ so $\mathrm{Cov}(\mathbf{y}|\mathbf{X},\mathbf{z},I^c) = \mathrm{Cov}(\mathbf{y}|\mathbf{X},\mathbf{z})$ and $\|\mathrm{Cov}(\mathbf{y}|\mathbf{X},\mathbf{z})\| \leqslant \sigma$. By an argument similar to the above discussion regarding (I), we have $\mathrm{E}\left[|\widehat{e}(\mathbf{X},\mathbf{z}) - e^*(\mathbf{X},\mathbf{z})|^2\,\Big|\,I^c\right] = o_P(1)$. Therefore, (II) is

also $o_P(1)$. For the third term, we have

$$
\begin{aligned}
\mathrm{P}\left(\|(\mathrm{III})\|_\infty \geqslant \epsilon \mid I^c\right) &\leqslant \sum_{j=1}^d \mathrm{P}\left(\sqrt{n}\left|\mathbb{P}_n \frac{w(\widehat{e}-e^*)}{\widehat{e}e^*}\mathbf{x}_{(j)}^\top(\widehat{\mathbf{g}}-\mathbf{g}^*)\right| \geqslant \epsilon \,\Big|\, I^c\right) \\
&\leqslant \sqrt{n}\sum_{j=1}^d \mathrm{E}\left[\left|\frac{w(\widehat{e}(\mathbf{X},\mathbf{z})-e^*(\mathbf{X},\mathbf{z}))}{\widehat{e}(\mathbf{X},\mathbf{z})e^*(\mathbf{X},\mathbf{z})}\mathbf{x}_{(j)}^\top(\widehat{\mathbf{g}}(\mathbf{X},\mathbf{z})-\mathbf{g}^*(\mathbf{X},\mathbf{z}))\right|\,\Big|\,I^c\right]\Big/\epsilon^2 \\
&\overset{(f)}{\leqslant} \frac{2\sqrt{n}}{\kappa^2}\sum_{j=1}^d \mathrm{E}\left[\left|(\widehat{e}(\mathbf{X},\mathbf{z})-e^*(\mathbf{X},\mathbf{z}))\,\mathbf{x}_{(j)}^\top(\widehat{\mathbf{g}}(\mathbf{X},\mathbf{z})-\mathbf{g}^*(\mathbf{X},\mathbf{z}))\right|\,\Big|\,I^c\right]\Big/\epsilon^2 + o_P(1) \\
&\overset{(g)}{\leqslant} \frac{2\sqrt{n}}{\kappa^2}\sum_{j=1}^d \mathrm{E}\left[|\widehat{e}(\mathbf{X},\mathbf{z})-e^*(\mathbf{X},\mathbf{z})|\,\|\mathbf{x}_{(j)}\|\,\|\widehat{\mathbf{g}}(\mathbf{X},\mathbf{z})-\mathbf{g}^*(\mathbf{X},\mathbf{z}))\|\,\Big|\,I^c\right]\Big/\epsilon^2 + o_P(1) \\
&\leqslant \frac{2\sqrt{n}C}{\kappa^2}\sum_{j=1}^d \mathrm{E}\left[|\widehat{e}(\mathbf{X},\mathbf{z})-e^*(\mathbf{X},\mathbf{z})|\,\|\widehat{\mathbf{g}}(\mathbf{X},\mathbf{z})-\mathbf{g}^*(\mathbf{X},\mathbf{z}))\|\,\Big|\,I^c\right]\Big/\epsilon^2 + o_P(1),
\end{aligned}
$$

where (f) follows from an argument similar to above and (g) follows from the Cauchy–Schwarz inequality. Finally, we have

$$
\mathrm{E}\left[|\widehat{e}(\mathbf{X},\mathbf{z})-e^*(\mathbf{X},\mathbf{z})|\,\|\widehat{\mathbf{g}}(\mathbf{X},\mathbf{z})-\mathbf{g}^*(\mathbf{X},\mathbf{z}))\|\right] \leqslant \|\widehat{e}(\mathbf{X},\mathbf{z})-e^*(\mathbf{X},\mathbf{z})\|_{P,2}\,\|\widehat{\mathbf{g}}(\mathbf{X},\mathbf{z})-\mathbf{g}^*(\mathbf{X},\mathbf{z})\|_{P,2}
$$
$$
= a(n)/n^{r_1+r_2} = o(n^{-1/2}).
$$

Therefore, $\mathrm{P}\left(\|(\mathrm{III})\|_\infty \geqslant \epsilon \mid I^c\right)$ is $o_P(1)$ and (III) is $o_P(1)$ too. We conclude the proof.

## A.4. Proof of Lemma A.2

For any functions $e(\cdot)$ and $\mathbf{g}(\cdot)$, data point $\Xi$, and $\boldsymbol{\beta}$, let us define

$$
\bar{\ell}(\Xi; e, \mathbf{g}; \boldsymbol{\beta}) := b(\mathbf{X}\boldsymbol{\beta}) - \left[\mathbf{g}(\mathbf{X},\mathbf{z}) + \frac{w}{e(\mathbf{X},\mathbf{z})}(\mathbf{g}(\mathbf{X},\mathbf{z})-\mathbf{y})\right]^\top \mathbf{X}\boldsymbol{\beta},
$$

so $\nabla_{\boldsymbol{\beta}}\bar{\ell}(\Xi; e, \mathbf{g}; \boldsymbol{\beta}) = \boldsymbol{\psi}(\Xi; e, \mathbf{g}; \boldsymbol{\beta})$ and $\nabla_{\boldsymbol{\beta}}^2\bar{\ell}(\Xi; e, \mathbf{g}; \boldsymbol{\beta}) = \mathbf{X}^\top \nabla b_{\boldsymbol{\theta}}(\mathbf{X}\boldsymbol{\beta})\mathbf{X} \succeq 0$. Therefore $\bar{\ell}(\cdot)$ is convex. In particular, $\mathrm{E}[\bar{\ell}(\Xi, e^*, \mathbf{g}^*, \boldsymbol{\beta})]$ is convex in $\boldsymbol{\beta}$. By Lemma A.1, we know that

$$
\nabla_{\boldsymbol{\beta}}P\bar{\ell}(e^*, \mathbf{g}^*, \boldsymbol{\beta}^*) = P\boldsymbol{\psi}(e^*, \mathbf{g}^*; \boldsymbol{\beta}^*) = 0.
$$

Also, for any fixed $\boldsymbol{\beta} \in \mathcal{B}$ it holds that $\nabla_{\boldsymbol{\beta}}^2 P\bar{\ell}(e^*, \mathbf{g}^*, \boldsymbol{\beta}) = P\mathbf{X}^\top \nabla_{\boldsymbol{\theta}}^2 b(\mathbf{X}\boldsymbol{\beta})\mathbf{X} \succ 0$. The last inequality follows because by assumption for any $\mathbf{X} \in \check{\mathcal{X}}$, $\lambda_{\min}\left(\nabla_{\boldsymbol{\theta}}^2 b(\mathbf{X}\boldsymbol{\beta})\right) > 0$. For the purpose of contradiction, if $\inf_{\mathbf{X} \in \check{\mathcal{X}}} \lambda_{\min}\left(\nabla_{\boldsymbol{\theta}}^2 b(\mathbf{X}\boldsymbol{\beta})\right) = 0$, there is a converging sequence $\{\mathbf{X}_m\}_{m=1}^\infty$ with limit $\widetilde{\mathbf{X}} \in \check{\mathcal{X}}$ such that $\lambda_{\min}\left(\nabla_{\boldsymbol{\theta}}^2 b(\widetilde{\mathbf{X}}\boldsymbol{\beta})\right) = 0$ by continuity of $\nabla_{\boldsymbol{\theta}}^2 b(\cdot)$. This is a contradiction. Therefore, $\nabla_{\boldsymbol{\beta}}^2 P\bar{\ell}(e^*, \mathbf{g}^*, \boldsymbol{\beta}) \succ \inf_{\mathbf{X} \in \check{\mathcal{X}}} \lambda_{\min}\left(\nabla_{\boldsymbol{\theta}}^2 b(\mathbf{X}\boldsymbol{\beta})\right) \cdot P\mathbf{X}^\top\mathbf{X} \succ 0$. In otherwords, $P\bar{\ell}(e^*, \mathbf{g}^*, \boldsymbol{\beta})$ is strictly convex. Therefore, $\boldsymbol{\beta}^*$ is the unique minimum. By compactness of $\check{\mathcal{X}}$ and continuity of $b(\cdot)$, we have $\mathbb{P}_n\bar{\ell}(e^*, \mathbf{g}^*; \boldsymbol{\beta}) \to_P P\bar{\ell}(e^*, \mathbf{g}^*; \boldsymbol{\beta})$ as an application of the weak law of large numbers. By Theorem 2.7 of (Newey & McFadden, 1994), it holds that there is a random sequence $\{\check{\boldsymbol{\beta}}_n\}_{n=1}^\infty$ that solves $\min_{\boldsymbol{\beta} \in \mathcal{B}} \mathbb{P}_n\bar{\ell}(e^*, \mathbf{g}^*; \boldsymbol{\beta})$ and converges to $\boldsymbol{\beta}^*$ with probability one. This implies that with probability one there is a sequence $\{\check{\boldsymbol{\beta}}_n\}_{n=1}^\infty$ such that $\mathbb{P}_n\boldsymbol{\psi}(e^*, \mathbf{g}^*; \check{\boldsymbol{\beta}}_n) = \mathbf{0}$. By Lemma A.1, it holds that

$$
\sqrt{n}\mathbb{P}_n\boldsymbol{\psi}(\widehat{e}, \widehat{\mathbf{g}}; \check{\boldsymbol{\beta}}_n) = \sqrt{n}\mathbb{P}_n\boldsymbol{\psi}(e^*, \mathbf{g}^*; \check{\boldsymbol{\beta}}_n) + \sqrt{n}\left(\mathbb{P}_n\boldsymbol{\tau}(e^*, \mathbf{g}^*) - \mathbb{P}_n\boldsymbol{\tau}(\widehat{e}, \widehat{\mathbf{g}})\right) = o_P(1), \tag{16}
$$

or $\inf_{\boldsymbol{\beta} \in \mathcal{B}} \left\|\mathbb{P}_n\boldsymbol{\psi}(\widehat{e}, \widehat{\mathbf{g}}; \boldsymbol{\beta})\right\| = o_P(1/\sqrt{n})$. This implies $\mathbb{P}_n\boldsymbol{\psi}(\widehat{e}, \widehat{\mathbf{g}}; \widehat{\boldsymbol{\beta}}) = o_P(1/\sqrt{n})$. Using the argument in (16) again, we have $\mathbb{P}_n\boldsymbol{\psi}(e^*, \mathbf{g}^*; \widehat{\boldsymbol{\beta}}) = o_P(1/\sqrt{n})$. This concludes the proof.

## A.5. Proof of Lemma A.3

The notations and definitions are the same as in the proof of the Lemma A.2. Fix arbitrary $\epsilon_1 < \epsilon_2$ such that $\overline{B(\boldsymbol{\beta}^*, \epsilon_2)} \subset \mathcal{B}$. By (Andersen & Gill, 1982), convexity and the compactness of $\overline{B(\boldsymbol{\beta}^*, \epsilon_2)}$, it holds that $\mathbb{P}_n\boldsymbol{\psi}(e^*, \mathbf{g}^*; \boldsymbol{\beta})$ converges to

$P\psi(e^*, \mathbf{g}^*; \boldsymbol{\beta})$ in probability uniformly on $\overline{B(\boldsymbol{\beta}^*, \epsilon_2)}$. Further, on $\overline{B(\boldsymbol{\beta}^*, \epsilon_2)} \setminus B(\boldsymbol{\beta}^*, \epsilon_1)$, by a Taylor expansion, it must be that

$$P\bar{\ell}(e^*, \mathbf{g}^*; \boldsymbol{\beta}) = P\bar{\ell}(e^*, \mathbf{g}^*; \boldsymbol{\beta}^*) + \frac{1}{2}(\boldsymbol{\beta} - \boldsymbol{\beta}^*)^\top P\mathbf{X}^\top \nabla_{\boldsymbol{\theta}}^2 b(\mathbf{X}\tilde{\boldsymbol{\beta}})\mathbf{X}(\boldsymbol{\beta} - \boldsymbol{\beta}^*) \overset{(a)}{\geqslant} P\bar{\ell}(e^*, \mathbf{g}^*; \boldsymbol{\beta}^*) + \frac{c}{2}\|\boldsymbol{\beta} - \boldsymbol{\beta}^*\|^2$$

which can be further lower bounded by $P\bar{\ell}(e^*, \mathbf{g}^*; \boldsymbol{\beta}^*) + \frac{c\epsilon_1^2}{2}$, for some $\tilde{\boldsymbol{\beta}} \in [\boldsymbol{\beta}^*, \boldsymbol{\beta}] \subset \overline{B(\boldsymbol{\beta}^*, \epsilon_2)}$ that depends on $\boldsymbol{\beta}$ and a universal $c > 0$. We prove the inequality (a) as follows. It suffices to show that $\lambda_{\min}\left(\nabla_{\boldsymbol{\theta}}^2 b(\mathbf{X}\boldsymbol{\beta})\right)$ is uniformly lower bounded by some positive number, when $\boldsymbol{\beta}$ ranges over $\overline{B(\boldsymbol{\beta}^*, \epsilon_2)}$ and $\mathbf{X}$ ranges over $\mathcal{X}$. Suppose otherwise, there is then $\{(\mathbf{X}_m, \boldsymbol{\beta}_m)\}_{m=1}^\infty$ such that $\lambda_{\min}\left(\nabla_{\boldsymbol{\theta}}^2 b(\mathbf{X}_m\boldsymbol{\beta}_m)\right) \to 0$. By taking subsequence is necessary, we assume that $\mathbf{X}_m \to \bar{\mathbf{X}} \in \check{\mathcal{X}}$ and $\boldsymbol{\beta}_m \to \bar{\boldsymbol{\beta}} \in \overline{B(\boldsymbol{\beta}^*, \epsilon_2)} \subset \mathcal{B}$. By assumption $\bar{\mathbf{X}}\bar{\boldsymbol{\beta}} \in \Theta$ and continuity of $\nabla^2 b(\cdot)$ implies $\lambda_{\min}\left(\nabla_{\boldsymbol{\theta}}^2 b(\bar{\mathbf{X}}\bar{\boldsymbol{\beta}})\right) = 0$, which contradicts our assumption.

Consequently, by the uniform convergence in probability on $\overline{B(\boldsymbol{\beta}^*, \epsilon_2)}$, with probability approaching one, it must be that $\forall \boldsymbol{\beta} \in \overline{B(\boldsymbol{\beta}^*, \epsilon_2)} \setminus B(\boldsymbol{\beta}^*, \epsilon_1)$,

$$\mathbb{P}_n\bar{\ell}(e^*, \mathbf{g}^*; \boldsymbol{\beta}) - \frac{c\epsilon_1^2}{4} \geqslant \mathbb{P}_n\bar{\ell}(e^*, \mathbf{g}^*; \boldsymbol{\beta}^*) \overset{(b)}{\geqslant} \mathbb{P}_n\bar{\ell}(e^*, \mathbf{g}^*; \boldsymbol{\beta}) + (\boldsymbol{\beta} - \boldsymbol{\beta}^*)^\top \mathbb{P}_n\psi(e^*, \mathbf{g}^*; \boldsymbol{\beta})$$

$$= \mathbb{P}_n\bar{\ell}(e^*, \mathbf{g}^*; \boldsymbol{\beta}) + \|\boldsymbol{\beta} - \boldsymbol{\beta}^*\| \cdot \frac{(\boldsymbol{\beta}^* - \boldsymbol{\beta})^\top}{\|\boldsymbol{\beta} - \boldsymbol{\beta}^*\|} \mathbb{P}_n\psi(e^*, \mathbf{g}^*; \boldsymbol{\beta}),$$

where (b) follows from subgradient inequality. This implies that

$$\frac{(\boldsymbol{\beta} - \boldsymbol{\beta}^*)^\top}{\|\boldsymbol{\beta} - \boldsymbol{\beta}^*\|} \mathbb{P}_n\psi(e^*, \mathbf{g}^*; \boldsymbol{\beta}) \geqslant \frac{c\epsilon_1^2}{4\epsilon_2}. \tag{17}$$

Next, consider any $\boldsymbol{\beta} \in \mathcal{B} \setminus \overline{B(\boldsymbol{\beta}^*, \epsilon_2)}$, there is $\tilde{\boldsymbol{\beta}} = \lambda\boldsymbol{\beta} + (1 - \lambda)\boldsymbol{\beta}^* \in \overline{B(\boldsymbol{\beta}^*, \epsilon_2)} \setminus B(\boldsymbol{\beta}^*, \epsilon_1)$. By the integral form of the intermediate value theorem,

$$\frac{(\boldsymbol{\beta} - \boldsymbol{\beta}^*)^\top}{\|\boldsymbol{\beta} - \boldsymbol{\beta}^*\|}\left(\mathbb{P}_n\psi(e^*, \mathbf{g}^*; \boldsymbol{\beta}) - \mathbb{P}_n\psi(e^*, \mathbf{g}^*; \boldsymbol{\beta}^*)\right)$$

$$= \frac{(\boldsymbol{\beta} - \boldsymbol{\beta}^*)^\top}{\|\boldsymbol{\beta} - \boldsymbol{\beta}^*\|} \int_0^1 \mathbb{P}_n\mathbf{X}^\top \nabla_{\boldsymbol{\theta}}^2 b\left(\mathbf{X}(\boldsymbol{\beta}^* + t(\boldsymbol{\beta} - \boldsymbol{\beta}^*))\right)\mathbf{X}dt(\boldsymbol{\beta} - \boldsymbol{\beta}^*)$$

$$= \frac{(\tilde{\boldsymbol{\beta}} - \boldsymbol{\beta}^*)^\top}{\lambda\|\tilde{\boldsymbol{\beta}} - \boldsymbol{\beta}^*\|} \int_0^1 \mathbb{P}_n\mathbf{X}^\top \nabla_{\boldsymbol{\theta}}^2 b\left(\mathbf{X}\left(\boldsymbol{\beta}^* + \frac{t}{\lambda}(\boldsymbol{\beta} - \boldsymbol{\beta}^*)\right)\right)\mathbf{X}dt(\tilde{\boldsymbol{\beta}} - \boldsymbol{\beta}^*)$$

$$\overset{(c)}{=} \frac{(\tilde{\boldsymbol{\beta}} - \boldsymbol{\beta}^*)^\top}{\|\tilde{\boldsymbol{\beta}} - \boldsymbol{\beta}^*\|} \int_0^{\frac{1}{\lambda}} \mathbb{P}_n\mathbf{X}^\top \nabla_{\boldsymbol{\theta}}^2 b\left(\mathbf{X}\left(\boldsymbol{\beta}^* + t(\boldsymbol{\beta} - \boldsymbol{\beta}^*)\right)\right)\mathbf{X}dt(\tilde{\boldsymbol{\beta}} - \boldsymbol{\beta}^*)$$

$$\overset{(d)}{\geqslant} \frac{(\tilde{\boldsymbol{\beta}} - \boldsymbol{\beta}^*)^\top}{\|\tilde{\boldsymbol{\beta}} - \boldsymbol{\beta}^*\|} \int_0^1 \mathbb{P}_n\mathbf{X}^\top \nabla_{\boldsymbol{\theta}}^2 b\left(\mathbf{X}\left(\boldsymbol{\beta}^* + t(\boldsymbol{\beta} - \boldsymbol{\beta}^*)\right)\right)\mathbf{X}dt(\tilde{\boldsymbol{\beta}} - \boldsymbol{\beta}^*) \cdot$$

$$= \frac{(\tilde{\boldsymbol{\beta}} - \boldsymbol{\beta}^*)^\top}{\|\tilde{\boldsymbol{\beta}} - \boldsymbol{\beta}^*\|}\left(\mathbb{P}_n\psi(e^*, \mathbf{g}^*; \tilde{\boldsymbol{\beta}}) - \mathbb{P}_n\psi(e^*, \mathbf{g}^*; \boldsymbol{\beta}^*)\right) \overset{(e)}{\geqslant} \frac{c\epsilon_1^2}{4\epsilon_2} - \frac{(\tilde{\boldsymbol{\beta}} - \boldsymbol{\beta}^*)^\top}{\|\tilde{\boldsymbol{\beta}} - \boldsymbol{\beta}^*\|}\mathbb{P}_n\psi(e^*, \mathbf{g}^*; \boldsymbol{\beta}^*),$$

where (c) follows from a change of variable in the integral, (d) follows because the $\nabla_{\boldsymbol{\theta}}^2 b(\cdot)$ is positive definite, and (e) uses (17). Combining this again with (17), we conclude that with probability converging to one,

$$\|\mathbb{P}_n\psi(e^*, \mathbf{g}^*; \boldsymbol{\beta})\| \geqslant \frac{(\boldsymbol{\beta} - \boldsymbol{\beta}^*)^\top}{\|\boldsymbol{\beta} - \boldsymbol{\beta}^*\|}\mathbb{P}_n\psi(e^*, \mathbf{g}^*; \boldsymbol{\beta}) \geqslant \frac{c\epsilon_1^2}{4\epsilon_2} \quad \forall \boldsymbol{\beta} \in \mathcal{B} \setminus B(\boldsymbol{\beta}^*, \epsilon_2).$$

Finally, we note that by Lemma A.2, $\left\|\mathbb{P}_n\psi\left(e^*, \mathbf{g}^*; \widehat{\boldsymbol{\beta}}\right)\right\| = o_P(1/\sqrt{n})$. Therefore, it must be that $\widehat{\boldsymbol{\beta}} \in B(\boldsymbol{\beta}^*, \epsilon_1)$ with probability converging to one. We finish the proof.

## B. An Example of Failure of Dominance

Generally, if $e(\mathbf{X}, \mathbf{z})$ is carefully constructed so that it is "advantageous" to ignore the data points with missing label, one can show the following negative result.

*Table 3.* Non-Constant Propensity: Conjoint Analysis with Coverage-Based Stratified Sampling

| Method | (a) MAPE (%) | | | | (b) 95% CI Coverage (%) | | | | (c) Avg CI Width | | | |
|---|---|---|---|---|---|---|---|---|---|---|---|---|
| | 50 | 100 | 150 | 200 | 50 | 100 | 150 | 200 | 50 | 100 | 150 | 200 |
| Primary | 34.62 | 23.39 | 19.40 | 17.34 | 97.45 | 94.73 | 93.82 | 90.36 | 2.44 | 1.52 | 1.20 | 1.01 |
| Naive | 48.20 | 45.04 | 41.70 | 38.42 | 25.27 | 27.64 | 30.55 | 33.09 | 0.56 | 0.55 | 0.54 | 0.53 |
| PPI | – | 41.04 | 31.87 | 27.22 | – | 98.00 | 96.18 | 93.27 | – | 2.64 | 1.94 | 1.59 |
| PPI++ | – | 28.42 | 22.65 | 19.69 | – | 90.91 | 90.91 | 88.55 | – | 1.51 | 1.18 | 1.01 |
| RePPI | 44.93 | 28.03 | 22.42 | 20.31 | 99.09 | 93.09 | 91.09 | 88.36 | 2.95 | 1.61 | 1.23 | 1.04 |
| PSPA | 48.66 | 28.67 | 22.36 | 19.58 | 91.45 | 91.27 | 91.45 | 88.73 | 2.49 | 1.53 | 1.20 | 1.02 |
| GMM | 56.00 | 29.61 | 22.89 | 19.83 | 83.82 | 89.27 | 89.27 | 87.82 | 2.38 | 1.48 | 1.17 | 1.00 |
| Classic-M | 80.33 | 34.36 | 25.52 | 21.54 | 87.64 | 91.27 | 91.45 | 90.00 | 3.60 | 1.80 | 1.34 | 1.11 |
| GAI | 16.92 | 16.63 | 16.31 | 16.11 | 99.82 | 99.27 | 96.73 | 92.73 | 2.40 | 1.53 | 1.20 | 1.03 |
| GAI (Emb) | 17.06 | 16.59 | 16.10 | 16.12 | 99.82 | 98.91 | 98.36 | 96.55 | 2.42 | 1.62 | 1.28 | 1.14 |

**Notes**: A "–" indicates that the method encounters numerical instability at $n_P = 50$. Column headers indicate $n_P$ values; $n_A = 1,000$ throughout.

*Example* 1. (**Failure of Dominance**) Consider a setting with canonical GLMs such that the GLM density is correctly specified. Assume that $k = 1$ and $b(\theta) = \frac{1}{2}\theta^2$. In this case, we write $\mathbf{X} = \mathbf{x}^\top \in \mathbb{R}^{1 \times d}$. We further assume that $\mathbf{x}$ is generated through a mixture distribution and there is $\tilde{w}$ such that $\tilde{w} = 1$ with probability $p$ and zero otherwise. Also, $\mathrm{E}[\mathbf{x}\mathbf{x}^\top \mid \tilde{w}] = \mathbf{I}$ for all $\tilde{w} \in \{0, 1\}$. Conditional on $\tilde{w}$, the supports of $\mathbf{x}$ are disjoint and we denote them by $\mathcal{X}_w$. We assume that $\mathrm{E}[y \mid \mathbf{x}] = \mathbf{x}^\top \boldsymbol{\beta}^*$, and $\mathrm{Var}(y \mid \mathbf{x}) = \sigma_w^2$ whenever $\mathbf{x} \in \mathcal{X}_w$. Set $z = y - \mathbf{x}^\top \boldsymbol{\beta}$ so $\mathbf{g}^*(\mathbf{x}, z) = y$. $e(\mathbf{x}, z) = 1$ if $\mathbf{x} \in \mathcal{X}_1$ and $e(\mathbf{x}, z) = \kappa$ if $\mathbf{x} \in \mathcal{X}_0$. Conditional on $\tilde{w}$, $w$ is independent of $(y, z)$. In the setup, the GAI estimator leads to the same asymptotics variance as the case where $y$ is fully observed, i.e., $(p\sigma_1^2 + (1 - p)\sigma_0^2)\mathbf{I}$. It is easy to see that $\widehat{\boldsymbol{\beta}}^{\mathrm{P}}$ obtained under the score function $w_i \nabla_{\boldsymbol{\beta}} \ell(\mathbf{x}_i, \mathbf{y}_i; \boldsymbol{\beta})$ is asymptotically normal with covariance given by

$$\mathrm{AVar}(\widehat{\boldsymbol{\beta}}^{\mathrm{P}}) = \frac{p\sigma_1^2 + (1 - p)\sigma_0^2 \kappa}{[p + (1 - p)\kappa]^2}\mathbf{I}$$
$$\prec \mathrm{AVar}(\widehat{\boldsymbol{\beta}}^{\mathrm{GAI}}) = (p\sigma_1^2 + (1 - p)\sigma_0^2)\mathbf{I},$$

which holds when $\sigma_1, \kappa \downarrow 0$.

## C. Non-Constant Propensity Experiment

The dominance result in Corollary 5.4 requires constant propensity (random labeling). To investigate GAI's empirical performance when this assumption is violated, we conduct an additional experiment on the vaccine conjoint data using stratified sampling, where the probability of obtaining a human label varies with vaccine coverage.

**Setup.** We partition respondents into two strata based on the vaccine coverage duration attribute (1-year vs. 5-year coverage). Primary data are sampled with stratum-specific probabilities: 30% of primary labels come from the 1-year coverage stratum and 70% from the 5-year coverage stratum, resulting in propensity values that vary across strata. We fix $n_A = 1,000$ auxiliary observations and vary $n_P \in \{50, 100, 150, 200\}$. For GAI and GAI (Emb), we estimate the propensity $\widehat{e}(\mathbf{X}, \mathbf{z})$ via cross-fitting using a decision tree. All other experimental details follow the conjoint analysis in Section 6.

**Results.** Table 3 reports MAPE, coverage, and CI width under non-constant propensity. Despite the violation of the constant-propensity assumption required for theoretical dominance, GAI and GAI (Emb) achieve the lowest MAPE (approximately 16–17%) across all sample sizes, substantially outperforming all benchmarks. GAI also maintains coverage around 95% nominal level, while most benchmarks exhibit undercoverage even at moderate sample sizes. CI widths for GAI are comparable to those of PPI++ and other benchmarks, confirming that GAI's coverage advantage is not driven by overly conservative intervals.

These results demonstrate that GAI's practical advantages extend beyond the constant-propensity setting covered by Corollary 5.4. Even when propensity varies substantially across strata, GAI continues to deliver the largest accuracy gains

*Table 4.* Ablation Study: GAI Variants for Conjoint Analysis

| Method | (a) MAPE (%) | | | | (b) 95% CI Coverage (%) | | | | (c) Avg CI Width | | | |
|---|---|---|---|---|---|---|---|---|---|---|---|---|
| | 50 | 100 | 150 | 200 | 50 | 100 | 150 | 200 | 50 | 100 | 150 | 200 |
| Primary | 32.02 | 25.27 | 19.67 | 19.01 | 97.82 | 92.73 | 92.55 | 88.00 | 2.43 | 1.51 | 1.19 | 1.01 |
| GAI ($g{=}z$) | 21.35 | 19.97 | 19.32 | 18.07 | 100.00 | 99.64 | 97.27 | 92.00 | 2.75 | 1.97 | 1.58 | 1.34 |
| GAI (learned $g$) | 16.86 | 17.52 | 15.97 | 16.64 | 99.82 | 97.09 | 94.73 | 90.91 | 1.94 | 1.36 | 1.11 | 0.96 |
| GAI ($g$+noise) | 17.49 | 18.27 | 16.99 | 17.50 | 99.82 | 98.00 | 96.36 | 94.18 | 2.33 | 1.58 | 1.30 | 1.12 |
| GAI (Emb) | 16.50 | 17.23 | 15.73 | 16.24 | 99.45 | 98.55 | 96.91 | 94.55 | 2.11 | 1.50 | 1.21 | 1.05 |
| GAI (Emb, $g$+noise) | 17.32 | 17.65 | 16.90 | 17.16 | 99.82 | 98.73 | 96.91 | 95.82 | 2.33 | 1.61 | 1.32 | 1.14 |

**Notes**: Column headers indicate $n_P$ values; $n_A = 1{,}000$ throughout. "learned $g$" is the default GAI estimator used in all other experiments.

and the best coverage among all methods considered.

## D. Ablation Studies

To isolate the contribution of each component of the GAI framework, we conduct ablation experiments on the vaccine conjoint data using the same experimental setup as Section 6. We compare the following GAI variants:

- **GAI** ($g{=}z$): Sets the nuisance function $\widehat{g}(\mathbf{X}, \mathbf{z}) = \mathbf{z}$ directly, bypassing learning.
- **GAI (learned** $g$): The default GAI estimator, which learns $\widehat{g}(\mathbf{X}, \mathbf{z})$ from primary data.
- **GAI ($g$+noise)**: Adds Gaussian noise to the learned $\widehat{g}$ to simulate nuisance estimation error.
- **GAI (Emb)**: Replaces discrete LLM labels $\mathbf{z}$ with high-dimensional text embeddings as auxiliary features.
- **GAI (Emb, $g$+noise)**: Combines embeddings with noisy nuisance estimation.

**Results.** Table 4 reports MAPE, coverage, and CI width for all GAI variants. Comparing GAI ($g{=}z$) to GAI (learned $g$) reveals that learning the nuisance function $g$ reduces MAPE, confirming that the learned nuisance function extracts substantially more information from auxiliary data than using raw AI outputs directly. GAI ($g$+noise) shows only a modest increase in MAPE relative to GAI (learned $g$), demonstrating robustness to nuisance estimation error, which is consistent with the Neyman orthogonality of the score function. GAI (Emb) achieves marginal further gains over GAI (learned $g$), suggesting that richer information from AI can provide additional information.

## E. Detailed Results of Experiments

### E.1. Vaccine Conjoint Analysis: Robustness to Auxiliary Generator and Prompting

To assess the robustness of our results to the choice of auxiliary data generator, we conduct additional analyses that vary both the underlying LLM (e.g., GPT-3.5, GPT-4, GPT-4o) and the prompting strategy (e.g., basic prompting, chain-of-thought, and few-shot variants). These auxiliary labels are generated by Wang et al. (2024). We do not revisit prompt engineering details here, as our goal is not to optimize prompt design, but rather to evaluate whether our inference framework remains stable across heterogeneous sources and qualities of AI-generated labels.

Tables 5–7 report MAPE, empirical coverage probabilities, and confidence interval widths across all auxiliary generators and prompting schemes. Across all configurations, GAI consistently achieves lower MAPE than primary-only estimation, naive pooling estimation, and PPI-based methods, often by a substantial margin. At the same time, GAI maintains nominal coverage and produces confidence intervals that are comparable to or tighter than those of competing approaches. Therefore, GAI reliably improves estimation accuracy without sacrificing inferential validity, which is robust to substantial variation in the auxiliary data. This robustness reflects the central design of our method—AI outputs enter estimation only through $\mathbf{g}(\mathbf{X}, \mathbf{z})$ and are not required to act as surrogate labels for the outcome. As a result, changes in the auxiliary generator do not compromise validity.

*Table 5.* Benchmark Comparison for Conjoint Analysis: MAPE (%)

| Model | Prompt | $n_P = 50$ | | | | | $n_P = 100$ | | | | |
|---|---|---|---|---|---|---|---|---|---|---|---|
| | | Primary | Naive | PPI | PPI++ | GAI | Primary | Naive | PPI | PPI++ | GAI |
| GPT-3.5-Turbo-0613 | Basic | 32.02 | 21.22 | - | - | 18.32 | 25.27 | 20.74 | - | - | 18.26 |
| | CoT | 32.02 | 48.83 | - | - | 17.23 | 25.27 | 45.89 | - | - | 17.61 |
| GPT-3.5-Turbo-0125 | Basic | 32.02 | 22.10 | - | - | 18.36 | 25.27 | 22.06 | - | - | 18.30 |
| | CoT | 32.02 | 42.98 | - | - | 17.48 | 25.27 | 39.97 | 50.55 | 30.41 | 17.92 |
| GPT-4 | Basic | 32.02 | 44.41 | - | - | 16.63 | 25.27 | 41.47 | - | - | 17.15 |
| | CoT | 32.02 | 50.14 | - | - | 16.79 | 25.27 | 47.75 | 48.56 | 29.40 | 17.37 |
| GPT-4o | Basic | 32.02 | 45.08 | - | - | 16.66 | 25.27 | 42.61 | 48.32 | 30.19 | 17.43 |
| | CoT | 32.02 | 48.52 | - | - | 16.86 | 25.27 | 45.75 | 44.57 | 29.69 | 17.52 |
| | FS | 32.02 | 41.74 | - | - | 16.14 | 25.27 | 39.83 | 43.10 | 29.73 | 17.00 |
| GPT-4o Fine-tuned | Basic | 32.02 | 35.20 | - | - | 14.83 | 25.27 | 33.23 | 48.17 | 30.17 | 15.40 |

| Model | Prompt | $n_P = 150$ | | | | | $n_P = 200$ | | | | |
|---|---|---|---|---|---|---|---|---|---|---|---|
| | | Primary | Naive | PPI | PPI++ | GAI | Primary | Naive | PPI | PPI++ | GAI |
| GPT-3.5-Turbo-0613 | Basic | 19.67 | 17.62 | - | - | 16.68 | 19.01 | 17.21 | - | - | 17.59 |
| | CoT | 19.67 | 42.73 | 39.02 | 22.60 | 16.35 | 19.01 | 40.10 | 35.08 | 21.20 | 17.22 |
| GPT-3.5-Turbo-0125 | Basic | 19.67 | 20.53 | 43.82 | 23.15 | 17.24 | 19.01 | 20.61 | 35.71 | 20.92 | 17.70 |
| | CoT | 19.67 | 37.40 | 36.11 | 23.07 | 16.64 | 19.01 | 35.17 | 29.95 | 20.96 | 17.22 |
| GPT-4 | Basic | 19.67 | 39.70 | 31.29 | 22.59 | 15.80 | 19.01 | 38.21 | 29.91 | 20.95 | 16.48 |
| | CoT | 19.67 | 46.00 | 33.54 | 22.62 | 15.96 | 19.01 | 43.11 | 29.21 | 20.96 | 16.64 |
| GPT-4o | Basic | 19.67 | 40.58 | 33.60 | 22.91 | 16.06 | 19.01 | 38.65 | 30.14 | 20.93 | 16.97 |
| | CoT | 19.67 | 43.33 | 32.96 | 22.96 | 15.97 | 19.01 | 40.29 | 29.21 | 21.00 | 16.64 |
| | FS | 19.67 | 38.09 | 31.35 | 22.45 | 15.58 | 19.01 | 36.51 | 30.31 | 21.19 | 16.50 |
| GPT-4o Fine-tuned | Basic | 19.67 | 32.14 | 32.93 | 22.54 | 14.05 | 19.01 | 30.32 | 30.21 | 20.94 | 15.38 |

**Notes**: When calculating MAPE for all methods, we add a small constant to the denominator for better illustration when the magnitude of parameters is very small. A "–" symbol indicates cases where the value exceeds 1,000, which mainly occur because PPI-based methods can suffer from singularity problems in small primary samples.

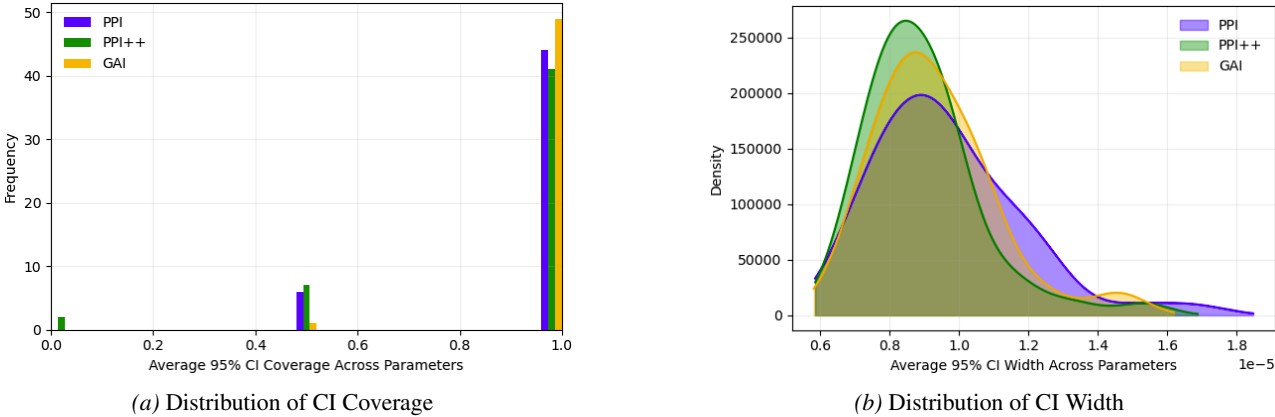

*(a)* Distribution of CI Coverage

*(b)* Distribution of CI Width

*Figure 4.* Comparison between PPI, PPI++ and GAI for Census Analysis ($n_P = 1000$)

### E.2. Census Analysis: Distribution of CI Coverage and Width

Figure 4 provides a complementary, distributional perspective on the census analysis results reported in Section 6.2. While the main tables summarize average performance across trials, these figures examine the full empirical distributions of inference diagnostics across repeated experiments.

Panel (a) plots the distribution of trial-level average 95% confidence interval coverage across parameters. All three methods—PPI, PPI++, and GAI—exhibit coverage distributions centered near the nominal level, indicating valid inference on average. This demonstrates that the efficiency gains delivered by GAI are not obtained by sacrificing coverage.

Panel (b) reports the distribution of average 95% confidence interval widths across parameters. The distributions for PPI,

*Table 6.* Benchmark Comparison for Conjoint Analysis: 95% CI Coverage Probability (%)

| Model | Prompt | $n_P = 50$ | | | | | $n_P = 100$ | | | | |
|---|---|---|---|---|---|---|---|---|---|---|---|
| | | Primary | Naive | PPI | PPI++ | GAI | Primary | Naive | PPI | PPI++ | GAI |
| GPT-3.5-Turbo-0613 | Basic | 97.82 | 69.27 | 100.00 | 78.55 | 100.00 | 92.73 | 67.82 | 100.00 | 86.00 | 95.64 |
| | CoT | 97.82 | 27.27 | 100.00 | 95.82 | 99.64 | 92.73 | 27.64 | 99.82 | 89.09 | 96.91 |
| GPT-3.5-Turbo-0125 | Basic | 97.82 | 38.73 | 100.00 | 96.73 | 100.00 | 92.73 | 41.09 | 99.82 | 88.91 | 96.73 |
| | CoT | 97.82 | 20.00 | 100.00 | 93.82 | 100.00 | 92.73 | 22.00 | 98.18 | 88.36 | 97.09 |
| GPT-4 | Basic | 97.82 | 44.55 | 100.00 | 92.55 | 99.45 | 92.73 | 45.09 | 98.18 | 89.64 | 97.45 |
| | CoT | 97.82 | 28.55 | 100.00 | 94.18 | 99.45 | 92.73 | 28.00 | 99.09 | 89.64 | 96.91 |
| | Basic | 97.82 | 28.36 | 100.00 | 94.18 | 99.45 | 92.73 | 31.09 | 97.64 | 88.91 | 97.09 |
| GPT-4o | CoT | 97.82 | 26.55 | 99.82 | 94.55 | 99.82 | 92.73 | 26.36 | 98.00 | 89.09 | 97.09 |
| | FS | 97.82 | 44.91 | 100.00 | 93.82 | 99.64 | 92.73 | 47.27 | 98.36 | 90.18 | 97.09 |
| GPT-4o Fine-tuned | Basic | 97.82 | 69.09 | 100.00 | 94.00 | 99.82 | 92.73 | 65.64 | 98.55 | 89.64 | 97.82 |

| Model | Prompt | $n_P = 150$ | | | | | $n_P = 200$ | | | | |
|---|---|---|---|---|---|---|---|---|---|---|---|
| | | Primary | Naive | PPI | PPI++ | GAI | Primary | Naive | PPI | PPI++ | GAI |
| GPT-3.5-Turbo-0613 | Basic | 92.55 | 76.73 | 100.00 | 89.45 | 92.18 | 88.00 | 74.36 | 100.00 | 88.00 | 87.09 |
| | CoT | 92.55 | 27.45 | 98.00 | 90.36 | 94.73 | 88.00 | 29.09 | 95.64 | 84.73 | 90.18 |
| GPT-3.5-Turbo-0125 | Basic | 92.55 | 44.00 | 98.91 | 89.82 | 92.36 | 88.00 | 45.09 | 97.45 | 84.36 | 87.45 |
| | CoT | 92.55 | 24.36 | 96.73 | 90.91 | 94.18 | 88.00 | 27.45 | 94.55 | 84.73 | 90.00 |
| GPT-4 | Basic | 92.55 | 45.09 | 96.55 | 90.36 | 96.18 | 88.00 | 45.64 | 92.73 | 84.36 | 91.82 |
| | CoT | 92.55 | 27.27 | 97.64 | 90.18 | 94.73 | 88.00 | 26.18 | 92.55 | 84.36 | 91.09 |
| | Basic | 92.55 | 33.64 | 96.55 | 89.82 | 95.09 | 88.00 | 36.00 | 92.00 | 84.36 | 90.91 |
| GPT-4o | CoT | 92.55 | 26.36 | 95.82 | 90.91 | 94.73 | 88.00 | 27.09 | 92.91 | 85.09 | 90.91 |
| | FS | 92.55 | 48.18 | 96.55 | 90.18 | 95.82 | 88.00 | 48.91 | 91.27 | 84.73 | 91.64 |
| GPT-4o Fine-tuned | Basic | 92.55 | 66.73 | 95.45 | 90.91 | 96.55 | 88.00 | 64.18 | 91.27 | 84.55 | 93.82 |

PPI++, and GAI are broadly comparable in magnitude, indicating that GAI does not rely on inflated uncertainty to achieve valid inference.

Taken together, these results support our theoretical guarantees and complement the mean-based comparisons in the main text by showing that improved performance is robust and not driven by a small number of favorable realizations.

### E.3. Tests for Performance Improvement

The main results in Section 6 report MAPEs, empirical coverage probabilities, and confidence interval widths across repeated trials. While these summaries clearly illustrate systematic differences across estimators, they do not directly assess whether the observed improvements of GAI over competing methods are statistically significant at the trial level. To complement the main tables and figures, we therefore conduct paired statistical tests that formally evaluate whether GAI delivers consistent performance improvements relative to benchmark estimators across identical experimental draws.

For each configuration of $(n_P, n_A)$ and for each performance metric (average MAPE across parameters, average coverage probability across parameters, and average confidence interval width across parameters), we perform paired $t$-tests comparing GAI to each benchmark method (Primary, Naive, PPI, PPI++, RePPI, PSPA, GMM, Classic-M). The tests are conducted across the repeated experimental trials, using the same underlying subsamples for all methods within each trial. This paired design controls variation driven by random sampling of observations and, thus, isolates estimator differences. Reported $p$-values correspond to the null hypothesis that the mean performance difference between GAI and the benchmark method is zero.

**Conjoint Analysis Results.** Table 8 reports paired $t$-test $p$-values for the vaccine conjoint analysis. Across all primary sample sizes, GAI achieves statistically significant improvements in MAPE relative to Primary, Naive, PPI++, PSPA, GMM, and Classic-M ($p < 0.01$). Improvements relative to PPI and RePPI are also statistically significant for moderate and large primary samples, while differences are not significant at $n_P = 50$. Coverage differences relative to all benchmarks, except for PPI, are statistically significant, indicating that GAI improves inferential validity. For confidence interval width, GAI yields significantly tighter intervals than all benchmarks in nearly all configurations.

Table 7. Benchmark Comparison for Conjoint Analysis: 95% CI Width

| | | $n_P = 50$ | | | | | $n_P = 100$ | | | | |
|---|---|---|---|---|---|---|---|---|---|---|---|
| Model | Prompt | Primary | Naive | PPI | PPI++ | GAI | Primary | Naive | PPI | PPI++ | GAI |
| GPT-3.5-Turbo-0613 | Basic | 2.43 | 0.88 | 115.88 | 2.08 | 1.89 | 1.51 | 0.80 | 39.78 | 1.43 | 1.35 |
| | CoT | 2.43 | 0.57 | 14.21 | 2.62 | 1.93 | 1.51 | 0.54 | 5.18 | 1.50 | 1.36 |
| GPT-3.5-Turbo-0125 | Basic | 2.43 | 0.44 | 16.54 | 2.37 | 1.91 | 1.51 | 0.43 | 4.87 | 1.49 | 1.35 |
| | CoT | 2.43 | 0.51 | 37.35 | 2.48 | 1.92 | 1.51 | 0.49 | 3.30 | 1.51 | 1.35 |
| GPT-4 | Basic | 2.43 | 0.54 | 11.43 | 2.43 | 1.93 | 1.51 | 0.52 | 2.88 | 1.52 | 1.35 |
| | CoT | 2.43 | 0.56 | 21.23 | 2.44 | 1.93 | 1.51 | 0.54 | 3.41 | 1.51 | 1.35 |
| GPT-4o | Basic | 2.43 | 0.54 | 14.68 | 2.49 | 1.93 | 1.51 | 0.52 | 3.06 | 1.51 | 1.35 |
| | CoT | 2.43 | 0.55 | 9.28 | 2.44 | 1.94 | 1.51 | 0.53 | 2.71 | 1.52 | 1.36 |
| | FS | 2.43 | 0.55 | 11.42 | 2.45 | 1.94 | 1.51 | 0.53 | 2.75 | 1.50 | 1.35 |
| GPT-4o Fine-tuned | Basic | 2.43 | 0.57 | 12.99 | 2.53 | 1.93 | 1.51 | 0.55 | 2.95 | 1.50 | 1.36 |

| | | $n_P = 150$ | | | | | $n_P = 200$ | | | | |
|---|---|---|---|---|---|---|---|---|---|---|---|
| Model | Prompt | Primary | Naive | PPI | PPI++ | GAI | Primary | Naive | PPI | PPI++ | GAI |
| GPT-3.5-Turbo-0613 | Basic | 1.19 | 0.73 | 45.78 | 1.15 | 1.10 | 1.01 | 0.68 | 24.98 | 0.97 | 0.96 |
| | CoT | 1.19 | 0.52 | 2.39 | 1.19 | 1.11 | 1.01 | 0.50 | 1.99 | 1.01 | 0.96 |
| GPT-3.5-Turbo-0125 | Basic | 1.19 | 0.42 | 2.75 | 1.18 | 1.10 | 1.01 | 0.41 | 2.08 | 1.01 | 0.95 |
| | CoT | 1.19 | 0.47 | 2.14 | 1.19 | 1.11 | 1.01 | 0.46 | 1.67 | 1.01 | 0.96 |
| GPT-4 | Basic | 1.19 | 0.50 | 1.88 | 1.18 | 1.11 | 1.01 | 0.49 | 1.59 | 1.00 | 0.96 |
| | CoT | 1.19 | 0.52 | 1.97 | 1.19 | 1.11 | 1.01 | 0.50 | 1.61 | 1.00 | 0.96 |
| GPT-4o | Basic | 1.19 | 0.50 | 1.97 | 1.18 | 1.11 | 1.01 | 0.48 | 1.63 | 1.00 | 0.96 |
| | CoT | 1.19 | 0.51 | 1.94 | 1.19 | 1.11 | 1.01 | 0.49 | 1.58 | 1.00 | 0.96 |
| | FS | 1.19 | 0.51 | 1.87 | 1.18 | 1.11 | 1.01 | 0.50 | 1.59 | 1.00 | 0.96 |
| GPT-4o Fine-tuned | Basic | 1.19 | 0.53 | 1.95 | 1.19 | 1.11 | 1.01 | 0.51 | 1.62 | 1.00 | 0.96 |

**Census Analysis Results.** Table 9 presents analogous tests for the census application with continuous and well-calibrated auxiliary predictions. GAI significantly outperforms all benchmark methods in MAPE across all primary sample sizes, including PPI, PPI++, RePPI, PSPA, GMM, and Classic-M. Coverage improvements over all benchmark methods are also significant at most sample sizes. Importantly, GAI attains these coverage levels with confidence interval widths that are comparable to those of PPI-based and other benchmark methods.

Taken together, these tests confirm that the empirical gains reported in Section 6 are not driven by a small number of favorable draws, but instead reflect systematic and statistically significant improvements delivered by GAI. The results further underscore a central message of the paper: by incorporating auxiliary AI outputs as informative features rather than surrogate labels, GAI achieves robust accuracy gains and efficient inference across heterogeneous auxiliary-data regimes, consistently outperforming all benchmark approaches—including PPI, PPI++, RePPI, PSPA, GMM, and Classic-M—without sacrificing coverage.

*Table 8.* Paired t-test p-values (GAI vs benchmarks) for Conjoint Analysis

| Method | $n_P$=50 | $n_P$=100 | $n_P$=150 | $n_P$=200 |
|---|---|---|---|---|
| **MAPE** | | | | |
| Primary | $3.18 \times 10^{-15}$ | $1.54 \times 10^{-12}$ | $4.65 \times 10^{-7}$ | $2.57 \times 10^{-6}$ |
| Naive | $5.77 \times 10^{-46}$ | $1.51 \times 10^{-44}$ | $5.31 \times 10^{-49}$ | $3.10 \times 10^{-40}$ |
| PPI | 0.32 | $2.08 \times 10^{-18}$ | $7.74 \times 10^{-15}$ | $1.41 \times 10^{-15}$ |
| PPI++ | 0.01 | $3.49 \times 10^{-15}$ | $2.89 \times 10^{-11}$ | $9.58 \times 10^{-11}$ |
| RePPI | 0.32 | 0.05 | $5.88 \times 10^{-17}$ | $8.58 \times 10^{-12}$ |
| PSPA | $4.56 \times 10^{-11}$ | $9.57 \times 10^{-16}$ | $4.81 \times 10^{-11}$ | $1.64 \times 10^{-10}$ |
| GMM | $6.22 \times 10^{-9}$ | $5.37 \times 10^{-19}$ | $2.84 \times 10^{-12}$ | $8.53 \times 10^{-12}$ |
| Classic-M | $7.70 \times 10^{-14}$ | $1.56 \times 10^{-16}$ | $1.97 \times 10^{-11}$ | $1.07 \times 10^{-12}$ |
| **95% CI Coverage Probability** | | | | |
| Primary | $2.74 \times 10^{-3}$ | $3.29 \times 10^{-3}$ | 0.08 | 0.12 |
| Naive | $1.15 \times 10^{-99}$ | $9.71 \times 10^{-93}$ | $1.95 \times 10^{-83}$ | $7.11 \times 10^{-64}$ |
| PPI | 1.00 | 0.29 | 0.33 | 0.24 |
| PPI++ | $1.83 \times 10^{-5}$ | $2.31 \times 10^{-6}$ | 0.01 | $2.03 \times 10^{-3}$ |
| RePPI | $1.67 \times 10^{-94}$ | $1.11 \times 10^{-3}$ | $7.15 \times 10^{-15}$ | $4.02 \times 10^{-8}$ |
| PSPA | $2.35 \times 10^{-6}$ | $3.93 \times 10^{-6}$ | 0.02 | $1.85 \times 10^{-3}$ |
| GMM | $3.72 \times 10^{-9}$ | $1.86 \times 10^{-8}$ | $1.46 \times 10^{-3}$ | $9.30 \times 10^{-4}$ |
| Classic-M | $2.02 \times 10^{-9}$ | $2.78 \times 10^{-5}$ | 0.01 | 0.02 |
| **95% CI Width** | | | | |
| Primary | $2.52 \times 10^{-27}$ | $2.91 \times 10^{-26}$ | $1.57 \times 10^{-24}$ | $1.06 \times 10^{-22}$ |
| Naive | $4.80 \times 10^{-53}$ | $1.53 \times 10^{-65}$ | $2.24 \times 10^{-70}$ | $5.24 \times 10^{-76}$ |
| PPI | 0.02 | $2.03 \times 10^{-22}$ | $3.51 \times 10^{-30}$ | $3.53 \times 10^{-32}$ |
| PPI++ | $2.94 \times 10^{-17}$ | $3.31 \times 10^{-20}$ | $4.43 \times 10^{-18}$ | $1.64 \times 10^{-18}$ |
| RePPI | $4.16 \times 10^{-60}$ | 0.32 | $3.48 \times 10^{-9}$ | $2.67 \times 10^{-7}$ |
| PSPA | $1.28 \times 10^{-19}$ | $3.12 \times 10^{-26}$ | $5.01 \times 10^{-22}$ | $5.50 \times 10^{-23}$ |
| GMM | 0.32 | $2.36 \times 10^{-17}$ | $1.76 \times 10^{-15}$ | $2.21 \times 10^{-14}$ |
| Classic-M | $7.78 \times 10^{-17}$ | $3.61 \times 10^{-32}$ | $1.46 \times 10^{-29}$ | $1.20 \times 10^{-36}$ |

*Table 9.* Paired t-test p-values (GAI vs benchmarks) for Census Analysis

| Method | $n_P$=100 | $n_P$=250 | $n_P$=500 | $n_P$=750 | $n_P$=1000 |
|---|---|---|---|---|---|
| **MAPE** | | | | | |
| Primary | $2.36\times10^{-10}$ | $2.17\times10^{-8}$ | $5.46\times10^{-9}$ | $4.57\times10^{-6}$ | $6.75\times10^{-6}$ |
| Naive | $1.23\times10^{-7}$ | $5.73\times10^{-8}$ | $2.55\times10^{-5}$ | $6.25\times10^{-4}$ | $4.07\times10^{-3}$ |
| PPI | $2.92\times10^{-7}$ | $6.70\times10^{-10}$ | $1.30\times10^{-7}$ | $5.62\times10^{-6}$ | $1.17\times10^{-4}$ |
| PPI++ | $1.43\times10^{-9}$ | $1.73\times10^{-9}$ | $1.25\times10^{-7}$ | $9.60\times10^{-6}$ | $1.59\times10^{-5}$ |
| RePPI | $8.73\times10^{-11}$ | $2.11\times10^{-10}$ | $6.99\times10^{-9}$ | $4.83\times10^{-6}$ | $1.24\times10^{-5}$ |
| PSPA | $4.43\times10^{-10}$ | $1.54\times10^{-9}$ | $2.95\times10^{-7}$ | $1.34\times10^{-5}$ | $2.57\times10^{-5}$ |
| GMM | $2.13\times10^{-10}$ | $6.53\times10^{-9}$ | $1.91\times10^{-6}$ | $2.37\times10^{-5}$ | $3.00\times10^{-5}$ |
| Classic-M | $1.00$ | $2.27\times10^{-8}$ | $1.87\times10^{-8}$ | $8.27\times10^{-6}$ | $3.95\times10^{-5}$ |
| **95% CI Coverage Probability** | | | | | |
| Primary | $3.09\times10^{-4}$ | $9.70\times10^{-5}$ | $9.70\times10^{-5}$ | $8.18\times10^{-4}$ | $1.28\times10^{-4}$ |
| Naive | $3.20\times10^{-25}$ | $4.00\times10^{-25}$ | $4.91\times10^{-25}$ | $6.31\times10^{-27}$ | $6.31\times10^{-27}$ |
| PPI | $1.92\times10^{-3}$ | $3.24\times10^{-2}$ | $2.38\times10^{-2}$ | $1.73\times10^{-1}$ | $5.20\times10^{-2}$ |
| PPI++ | $6.30\times10^{-4}$ | $1.92\times10^{-3}$ | $1.11\times10^{-2}$ | $5.22\times10^{-2}$ | $9.42\times10^{-3}$ |
| RePPI | $9.70\times10^{-5}$ | $3.48\times10^{-3}$ | $3.63\times10^{-3}$ | $6.64\times10^{-2}$ | $4.12\times10^{-2}$ |
| PSPA | $1.02\times10^{-3}$ | $2.05\times10^{-3}$ | $1.11\times10^{-2}$ | $4.12\times10^{-2}$ | $1.50\times10^{-2}$ |
| GMM | $1.02\times10^{-3}$ | $1.15\times10^{-3}$ | $1.11\times10^{-2}$ | $6.64\times10^{-2}$ | $2.51\times10^{-2}$ |
| Classic-M | $6.35\times10^{-3}$ | $1.68\times10^{-2}$ | $1.93\times10^{-2}$ | $1.05\times10^{-1}$ | $7.85\times10^{-2}$ |
| **95% CI Width** | | | | | |
| Primary | $8.57\times10^{-2}$ | $5.00\times10^{-5}$ | $3.71\times10^{-8}$ | $1.11\times10^{-13}$ | $1.15\times10^{-15}$ |
| Naive | $3.69\times10^{-10}$ | $1.25\times10^{-15}$ | $1.90\times10^{-18}$ | $3.51\times10^{-23}$ | $1.20\times10^{-24}$ |
| PPI | $7.78\times10^{-1}$ | $7.42\times10^{-1}$ | $9.78\times10^{-1}$ | $6.52\times10^{-1}$ | $3.60\times10^{-1}$ |
| PPI++ | $1.56\times10^{-1}$ | $5.42\times10^{-1}$ | $5.28\times10^{-1}$ | $2.55\times10^{-1}$ | $3.02\times10^{-1}$ |
| RePPI | $4.35\times10^{-3}$ | $3.79\times10^{-1}$ | $8.00\times10^{-1}$ | $7.59\times10^{-1}$ | $7.13\times10^{-1}$ |
| PSPA | $1.23\times10^{-2}$ | $4.58\times10^{-2}$ | $7.62\times10^{-2}$ | $7.32\times10^{-2}$ | $1.40\times10^{-1}$ |
| GMM | $9.34\times10^{-3}$ | $5.78\times10^{-2}$ | $1.09\times10^{-1}$ | $8.68\times10^{-2}$ | $1.31\times10^{-1}$ |
| Classic-M | – | $9.92\times10^{-1}$ | $4.80\times10^{-1}$ | $3.31\times10^{-1}$ | $1.08\times10^{-1}$ |

**Notes**: A "–" indicates numerical instability at the given $n_P$.

