# OpenReview forum: "Generative Augmented Inference"
_ICML.cc/2026/Conference — ICML 2026 regular_

### Official Review · Reviewer_tk46 · 2026-02-24

**Soundness:** 3
**Presentation:** 3
**Significance:** 3
**Originality:** 2
**Overall Recommendation:** 4
**Confidence:** 3

**Summary:**

This paper proposes a Neyman-orthogonal framework, termed GAI, for AI-augmented statistical inference by treating AI-generated representations as auxiliary predictive features rather than surrogate labels. By embedding these representations into a doubly robust score, the method enables valid √n inference under partial labeling and flexible machine learning estimation of nuisance components. Theoretically, the authors establish asymptotic normality and show that, under random labeling and when AI representations provide incremental information, GAI strictly dominates primary-only estimation in asymptotic variance. Empirically, the method demonstrates improved estimation accuracy and valid coverage across both low-quality discrete LLM outputs and high-quality probabilistic auxiliary predictions.

**Compliance With Llm Reviewing Policy:**

Affirmed.

**Key Questions For Authors:**

1 The idea of leveraging auxiliary variables to improve estimation efficiency has long existed in statistics and econometrics, such as in Doubly Robust Estimation. The only difference here is that the auxiliary variables are AI-generated representations.

2 The theoretical advantage of this paper over the primary-only estimator relies on the assumption of random labeling and identical distributions between the primary and auxiliary samples. This is a strong assumption and its limitations should be discussed.

3 GAI employs a more complex first-stage model and benefits from more flexible machine learning modeling capacity. Therefore, the comparison with PPI is not entirely fair.

4 The paper lacks ablation studies.

5 In the numerical experiments, the dimensionality of the AI representations is very low, and the experiments do not adequately reflect representation complexity.

**Limitations:**

Yes

**Strengths And Weaknesses:**

Strengths：
1 The paper introduces a principled shift from treating AI outputs as surrogate labels to viewing them as auxiliary predictive features, which improves AI-augmented inference compared with baseline PPI method.

2 This paper provides rigorous proofs of asymptotic normality and a strict variance dominance result under random labeling when AI representations provide incremental information.

3 Numerical results and comparisons are presented to illustrate its advantages.

Limitations：

1 The idea of leveraging auxiliary variables to improve estimation efficiency has long existed in statistics and econometrics, such as in Doubly Robust Estimation. The only difference here is that the auxiliary variables are AI-generated representations.

2 The theoretical advantage of this paper over the primary-only estimator relies on the assumption of random labeling and identical distributions between the primary and auxiliary samples. This is a strong assumption and its limitations should be discussed.

3 GAI employs a more complex first-stage model and benefits from more flexible machine learning modeling capacity. Therefore, the comparison with PPI is not entirely fair.

4 The paper lacks ablation studies.

5 In the numerical experiments, the dimensionality of the AI representations is very low, and the experiments do not adequately reflect representation complexity.

---

> ### Author Rebuttal · Authors · 2026-03-29
>
> We thank you for the positive assessment and constructive suggestions. We are encouraged that you see the value of the framework. We will revise the paper accordingly.
>
> **1. Relation to classical auxiliary-variable / doubly robust ideas.**
>
> We agree that using auxiliary variables to improve efficiency is a classical idea. However, the capability of modern LLMs to generate a wide range of informed data—especially in digital-twin and synthetic-response settings—makes a comprehensive, statistically rigorous framework for incorporating such outputs critically important. Existing methods in the PPI literature approach this problem too narrowly by treating AI outputs only as surrogate labels. In practice, AI can provide high-dimensional, unstructured features (e.g., chain-of-thought reasoning text, embeddings) that do not share the format of the outcome. Our method is designed specifically to handle this broader class of auxiliary information within a missing-outcome framework, while preserving valid inference through a Neyman-orthogonal score. We put everything into the model DML framework so simple cross-fitting guarantees valid inference for a large class of models (GLM). Importantly, in settings where existing methods like RePPI do apply, we can theoretically prove that GAI achieves exactly the same or but asymptotic efficiency (as detailed in our response to Reviewer ZoNz). When z is not a surrogate label, PPI-based methods cannot be applied at all, whereas GAI remains valid and provides a "safe default"—under random labeling, it can never worsen performance relative to ignoring AI data, regardless of how biased or inaccurate the AI outputs may be.
>
> **2. Assumptions behind the theoretical advantage.**
>
> We agree this should be stated more explicitly. Our asymptotic dominance result over primary-only estimation is proved only under random labeling with constant propensity, and thus matched primary and auxiliary distributions. We will state these assumptions more prominently and note the appendix limitation: when labeling propensity is non-constant, the clean dominance result need not hold.
>
> At the same time, we have now added a conjoint experiment with non-constant labeling propensity, where vaccines with longer coverage are more likely to receive human labels and e(X,z) must be estimated. Although the formal dominance guarantee no longer applies, GAI still outperforms the benchmark methods (as shown in the second section of our response to Reviewer Zbo8). We will add and discuss these results explicitly.
>
> **3. Comparison with PPI and the role of model flexibility.**
>
> We appreciate this point. Our claim is not that GAI wins despite using the same first stage as PPI. Rather, one advantage of GAI is that it can benefit from a richer, more flexible first-stage model. GAI uses AI outputs as auxiliary information and then corrects first-stage bias through the orthogonal score. This lets it exploit complex predictive structure in z while still delivering valid inference for the target parameter. We will revise the discussion to make clear that gains relative to PPI come from both the richer information set and the debiasing construction.
>
> **4. Ablation studies.**
>
> We agree and have now added ablations to isolate the sources of improvement. Specifically, we compare:
> (i) directly using g=z versus learning g(X,z), showing the value of learning rather than treating the AI output as a fixed proxy;
> (ii) learned g versus noisy learned g, showing robustness to nuisance estimation error; and
> (iii) label-based versus embedding-based auxiliary information, showing the value of retaining richer reasoning text rather than only low-dimensional labels.
>
> We will highlight these ablations more clearly in the revision.
>
> | Method                        | MAPE (%) | CI Coverage (%) | Avg CI Width |
> |------------------------------|---------:|----------------:|-------------:|
> | GAI (Label; g=z)         | 19.68    | 97.23           | 1.91         |
> | GAI (Label; learned g)   | 16.75    | 95.64           | 1.34         |
> | GAI (Label; noisy g)     | 17.56    | 97.09           | 1.58         |
> | GAI (Emb; learned g)     | 16.43    | 97.36           | 1.47         |
> | GAI (Emb; noisy g)       | 17.26    | 97.82           | 1.60         |
>
> **5. Representation complexity.**
>
> We agree that the original draft did not sufficiently emphasize richer AI representations. We have now added an embedding-based GAI variant in the conjoint experiment, where reasoning text generated by GPT-4o with CoT is converted into high-dimensional embeddings and used as auxiliary information. Empirically, the embedding-based variant achieves lower MAPE and better coverage than label-based GAI (as shown in our response to Reviewer ZoNz), confirming that the framework benefits from richer AI representations. We will revise the discussion to make this more explicit.
>
> Overall, we thank you for the helpful suggestions. They have helped us sharpen the paper.

---

### Official Review · Reviewer_dmw3 · 2026-03-11

**Soundness:** 3
**Presentation:** 3
**Significance:** 2
**Originality:** 2
**Overall Recommendation:** 3
**Confidence:** 4

**Summary:**

This paper studies the problem of valid statistical inference leveraging imperfect AI outputs. Instead of viewing the AI outputs as labels/surrogates, the authors propose to view the AI outputs as "features" or any side information that can improve the estimation and inference. Based on a Neyman-orthogonal score function, the authors prove asymptotic normality of the proposed estimator. The proposed methods are demonstrated on two case studies.

**Compliance With Llm Reviewing Policy:**

Affirmed.

**Key Questions For Authors:**

1. Could you clarify the novelty of the proposed method?
2. Is there some optimality (e.g., semiparametric efficiency) attached to the proposed method?
3. In the real data case studies, how is the ground-truth $\beta^*$ determined? Did you use a semi-synthetic setting?

**Limitations:**

yes

**Strengths And Weaknesses:**

Strength:

1. The problem studied is important.
2. The paper is clearly written and easy to follow.
3. The case studies are detailed.

Weakness:

1. The novelty of the proposed method is unclear; it feels more like a perspective change (or story re-telling) than a paradigm shift. In the PPI literature (or technically, classical semiparametric inference literature), using generated AI outputs as part of the features is already studied, e.g., in [1]. Adding a propensity score wouldn't be that surprising given these developments and classical semiparametric theory.

[1] Miao, Jiacheng, et al. "Assumption-lean and data-adaptive post-prediction inference." Journal of Machine Learning Research 26.179 (2025): 1-31.

---

> ### Author Rebuttal · Authors · 2026-03-29
>
> We thank you for the thoughtful feedback. We are encouraged that you find the problem important and the paper clearly written. We agree that the draft should better clarify our novelty relative to recent work, and will tone down language such as “paradigm shift” in the revision.
>
> **1. Novelty relative to Miao et al. (2025).**
>
> We appreciate this comparison. The difference between GAI and PSPA is not simply that we “add a propensity score.” The key distinction is where the prediction function enters the procedure. In PSPA, the prediction rule is assumed to be external; if we use an external LLM prediction in that framework, the LLM output serves as a proxy/prediction of the outcome, not part of the features. By contrast, our setting treats the AI output z itself as auxiliary information that may be categorical, biased, high-dimensional, or unstructured, and we first learn g(X,z) = E[Y|X,z] from labeled data. We then construct a Neyman-orthogonal score that corrects the bias induced by first-stage estimation. Thus, our contribution is not only “features instead of proxies,” but a debiased estimator for using AI outputs as informative features under missing outcomes. We believe this covers a much wider range of use cases for data augmentation with LLMs.
>
> We now include **PSPA** as a baseline in both studies. When implementing PSPA, we use the AI prediction as the external surrogate and compute the weight that optimally combines the primary score with a prediction-based correction (implemented in Python based on the paper and the authors’ R code). In both experiments, GAI achieves substantially lower MAPE than PSPA.
>
> Comparison to Benchmarks (Conjoint Analysis):
>
> | Method            | MAPE (%) | CI Coverage (%) | Avg CI Width |
> |-------------------|---------:|----------------:|-------------:|
> | Primary           | 23.99    | 92.77           | 1.53         |
> | PSPA              | 30.82    | 90.14           | 1.55         |
> | GAI               | 16.75    | 95.64           | 1.34         |
>
> Comparison to Benchmarks (Census Analysis):
>
> | Estimator         | MAPE   | CI Coverage | CI Width  |
> |-------------------|-------:|------------:|----------:|
> | Primary           | 527.43 | 79.2        | 1.16e-05  |
> | PSPA              | 446.65 | 88.2        | 1.30e-05  |
> | GAI               | 147.56 | 99.6        | 1.54e-05  |
>
> **2. Optimality / efficiency.**
>
> We do not claim that GAI is fully semiparametrically efficient for the DGP. Our theoretical guarantee is more targeted: under random labeling, Corollary 5.4 shows that GAI weakly dominates the primary-only estimator in asymptotic variance, with strict improvement whenever the AI representation is informative. When compared against PPI-based approaches in settings where they apply, we can show similar efficiency gains. However, as discussed above, GAI covers more general settings where PPI-based methods do not apply. Under non-random labeling (e depends on both X and z), although we cannot provide a theoretical efficiency guarantee, GAI still outperforms benchmarks in experiments (as shown in our response to Reviewer Zbo8). Moreover, when z is not a label—as is the case with free-form text or embeddings—PPI-based methods and PSPA cannot even be applied, whereas GAI remains asymptotically normal.
>
> **3. Ground-truth.**
>
> Our experiments use real datasets. Estimator accuracy is benchmarked against a “full-data” estimate obtained by fitting the same target model to the complete human-labeled dataset. In the conjoint study, ground-truth comes from fitting the logistic choice model to 4,605 fully labeled observations; in the census study, from fitting logistic regression to all 318,215 observations. Repeated trials subsample primary and auxiliary sets from these real data and compare each method’s estimate to this full-data benchmark. All ground-truth data are real, not semi-synthetic. We will clarify this in the revision.
>
> Overall, we appreciate your suggestions. They have helped us sharpen the paper’s positioning and we will clarify the distinction from methods such as PSPA, state our efficiency claim more precisely, and better explain the ground-truth construction. We hope you would agree that our method is more general than existing approaches in the literature.

---

> > ### Author Rebuttal · Reviewer_dmw3 · 2026-04-03
> >
> > Thank you for the clarification! My major concern is still about novelty so I'll maintain my score.

---

> > > ### Author Response · Authors · 2026-04-04
> > >
> > > In our rebuttal, we clarified that the contribution of GAI extends beyond simply incorporating a propensity score into the prediction-powered framework. To address your remaining concern of novelty we would like to further elaborate on an important distinction as follows:
> > >
> > > * ⁠In prediction-powered frameworks, such as the PSPA approach, the auxiliary information Z consists of variables external to the feature variables X, which are used to train an AI predictor $\hat{f}$ of the true human label. The learned predictor $\hat{f}$ is then used to augment the observed data.
> > > * ⁠⁠In contrast, our work addresses a fundamentally more general problem. The auxiliary information Z in our paper is the AI system’s output itself—this may include predicted labels, reasoning traces, or other unstructured data formats. Accordingly, we propose a new framework that goes beyond just label prediction — we directly models the relationship between these unstructured AI outputs and human labels. This enables us to construct a Neyman-orthogonal score that corrects for estimation error while leveraging the richness of AI-generated information. Therefore, PPI-based methods such as PASA only cover special cases of our applications.
> > > * ⁠Importantly, in settings where PPI-based methods and ours both apply, we can now also show that our estimator is strictly more efficient than the PPI-based estimators, including PSPA.
> > >
> > > We also provide empirical evidence demonstrating the improved performance of our approach. In the rebuttal, we included PSPA as a baseline—along with several additional benchmarks introduced in response to other reviewers—and found that GAI consistently outperforms these classical methods across our applications. We appreciate your suggestion of adding this baseline, and we will include them in the final version of the paper, together with the a discussion in the literature review to clarify the novelty.
> > >
> > > If our distinction has not yet been sufficiently clear, we would greatly appreciate any specific feedback on which aspects of our novelty claim remain unconvincing.
> > >
> > > We sincerely thank you for your feedback, which will help us sharpen the paper’s positioning and further improve the final version.

---

### Official Review · Reviewer_Zbo8 · 2026-03-13

**Soundness:** 3
**Presentation:** 3
**Significance:** 4
**Originality:** 2
**Overall Recommendation:** 4
**Confidence:** 4

**Summary:**

This paper introduces Generative Augmented Inference (GAI), a framework that integrates AI-generated outputs as auxiliary features to boost the efficiency of GLM estimation when human-labeled data is scarce. Instead of treating AI predictions as surrogate labels (which requires them to be somewhat accurate or calibrated), the authors treat them as informative signals within a Neyman-orthogonal, doubly robust score. This allows the method to remain valid even if the AI outputs are low-quality, while providing asymptotic variance reduction when they are informative. The authors establish asymptotic normality and show that GAI dominates human-only estimators under MCAR settings. Empirical results on LLM-annotated conjoint data and census data show improved point estimation and CI properties compared to PPI and standard baselines.

**Compliance With Llm Reviewing Policy:**

Affirmed.

**Key Questions For Authors:**

All experiments assume a constant e(X,z). How does GAI perform when the labeling mechanism is non-constant (MAR) and e must be estimated? Does the efficiency gain still hold up under misspecification?
How does GAI differ from classical semisupervised M-estimators that use unlabeled data? Adding a baseline from that literature would help clarify the specific value-add of the GAI framing.

**Limitations:**

The authors adequately discuss technical limitations, but could improve the paper by discussing potential negative societal impacts. For example, if auxiliary AI models contain demographic biases, the resulting GAI estimator could "tighten" around biased values in small-sample regimes. Suggesting subgroup audits as a guardrail would strengthen the limitations section.

**Strengths And Weaknesses:**

Soundness: The core estimator is technically sound and well-grounded in semiparametric theory. The use of a doubly robust, Neyman-orthogonal score is an appropriate choice for this problem, ensuring that the inference remains valid even if the nuisance models are not perfect. However, the theoretical "dominance" guarantee relies on the propensity score being a known constant (MCAR). In practice, labeling is rarely purely random, and the paper's own appendix shows that this dominance can fail when e(X,z) is non-constant. Additionally, the proofs rely on Donsker class assumptions, modern DML papers typically use cross-fitting to bypass these and allow for more complex, high-dimensional nuisance estimators.

Presentation: The paper is generally well-structured and the narrative is easy to follow. There is also a slight disconnect between the theory and the experiments: Algorithm 1 suggests using all data to fit nuisance functions, but the experimental section implies that g(X,z) was fit only on the primary data. Addressing these discrepancies would improve reproducibility.

Significance: This work addresses a highly relevant and timely problem. As LLMs become cheaper, many researchers are attempting to use AI labels to supplement small human-labeled sets. By providing a safe, principled framework that doesn't rely on the AI being a "perfect proxy," this work offers a valuable tool for practitioners. The experimental results, particularly the conjoint study using noisy LLM signals, demonstrate clear practical utility over existing methods like PPI.

Originality: The primary originality lies in the conceptual reframing: moving away from the "AI as a surrogate label" view and instead treating AI outputs as "auxiliary features" in a missing-outcome framework. While the mathematical machinery (AIPW/DML) is quite standard in the missing-data literature, its specific application to the generative AI pipeline is fresh and insightful. It provides a more robust alternative to recent "prediction-powered" approaches by removing the need for surrogate calibration.

---

> ### Author Rebuttal · Authors · 2026-03-29
>
> We thank you for the helpful suggestions and are encouraged by your positive assessment of the paper’s technical core and relevance. We will revise the paper accordingly.
>
> **1. Scope of the efficiency guarantee under constant e(X,z).**
>
> We agree this point should be stated more clearly. Our dominance result is proved only under random labeling with constant propensity. In that setting, Corollary 5.4 shows that GAI weakly dominates primary-only estimation in asymptotic variance, with strict improvement whenever the AI representation is informative. When e(X,z) varies with covariates, this guarantee need not hold. As noted in Appendix B, we provide a counterexample showing that dominance can fail when e(X,z) is designed so that ignoring unlabeled data is optimal.
>
> **2. Performance under non-constant e(X,z).**
>
> We have now run a conjoint variant with non-constant labeling propensity, where vaccines with longer coverage are more likely to receive human labels and e(X,z) must be estimated. Although the formal dominance guarantee no longer applies, GAI still outperforms all benchmarks. We will add these results to the revision.
>
> Comparison to Benchmarks (Conjoint Analysis with Non-constant Labeling Propensity):
>
> | Method            | MAPE (%) | CI Coverage (%) | Avg CI Width |
> |-------------------|---------:|----------------:|-------------:|
> | Primary           | 23.69    | 94.09           | 1.54         |
> | Naive             | 43.34    | 29.14           | 0.54         |
> | PPI               | 33.38    | 96.86           | 3.78         |
> | PPI++             | 23.59    | 90.67           | 1.71         |
> | Classic-M (w/o z) | 40.41    | 90.09           | 1.96         |
> | Classic-M (w/ z)  | 40.44    | 90.09           | 1.96         |
> | GAI               | 16.49    | 97.23           | 1.47         |
>
>
> **3. Donsker assumptions and cross-fitting.**
>
> To clarify, our proof mentions the Donsker class purely as an intermediate analysis step; it is not required as an assumption. Our implementation uses cross-fitting and does not rely on Donsker restrictions. We will revise the discussion to make this explicit.
>
> **4. Clarifying Algorithm 1.**
>
> We agree that Algorithm 1 is currently unclear and may suggest that both primary and auxiliary data are used to fit g(X,z). In fact, g(X,z) is fit using only the primary sample, since it requires human labels. We will revise Algorithm 1 and the text to make this explicit.
>
> **5. Comparison with classical semi-supervised M-estimators.**
>
> A key difference is that GAI explicitly corrects bias from first-stage estimation through a Neyman-orthogonal score, whereas classical semi-supervised M-estimators do not target this debiasing step. This distinction is especially important when the auxiliary signal comes from flexible, high-dimensional AI outputs where first-stage error may be substantial. We have added a benchmark based on Song, Lin, and Zhou (2024), reported as Classic-M, which reweights labeled observations using projection-based optimal weights with polynomial basis functions (implemented ourselves; no public code). GAI outperforms Classic-M in both studies. We will discuss this comparison more explicitly in the revision.
>
> Comparison to Benchmarks (Conjoint Analysis):
>
> | Method            | MAPE (%) | CI Coverage (%) | Avg CI Width |
> |-------------------|---------:|----------------:|-------------:|
> | Primary           | 23.99    | 92.77           | 1.53         |
> | Classic-M (w/o z) | 38.93    | 89.23           | 1.90         |
> | Classic-M (w/ z)  | 39.46    | 88.91           | 1.92         |
> | GAI               | 16.75    | 95.64           | 1.34         |
>
> Comparison to Benchmarks (Census Analysis):
>
> | Estimator         | MAPE   | CI Coverage | CI Width  |
> |-------------------|-------:|------------:|----------:|
> | Primary           | 527.43 | 79.2        | 1.16e-05  |
> | Classic-M (w/o z) | 411.36 | 89.0        | 1.51e-05  |
> | Classic-M (w/ z)  | 375.83 | 91.4        | 1.30e-05  |
> | GAI               | 147.56 | 99.6        | 1.54e-05  |
>
> **6. Subgroup Audits.**
>
> We will also strengthen the limitations section by recommending subgroup-level audits before deployment in sensitive settings.
>
> Overall, we thank you for these suggestions. They help us sharpen the paper with evidence under more realistic labeling mechanisms, improved clarity between theory and implementation, and better positioning relative to classical semi-supervised estimation.

---

> > ### Author Rebuttal · Reviewer_Zbo8 · 2026-04-03
> >
> > i think most of my questions\suggestions require major revision of the paper

---

> > > ### Author Response · Authors · 2026-04-04
> > >
> > > Thank you again for your thoughtful suggestions.
> > >
> > > In our rebuttal, we tried to address your concerns by providing additional clarification, new experimental results, and additional benchmarks. We will incorporate all of these changes into the final version of the paper, including a clearer scope of the efficiency guarantee, the new non-constant-propensity experiment, the revised algorithm with clearer cross-fitting implementation, and the expanded discussion and empirical comparison with classical semi-supervised methods.
> > >
> > > We hope these revisions help address your concerns and sincerely appreciate your feedback.

---

### Official Review · Reviewer_ZoNz · 2026-03-24

**Soundness:** 3
**Presentation:** 3
**Significance:** 2
**Originality:** 2
**Overall Recommendation:** 5
**Confidence:** 4

**Summary:**

This paper tackles the setting of using AI outputs as additional features in performing statistical estimation in the limited human data regime. The paper proposes a method for GLMs that uses a Neyman-orthogonal score with nuisance functions for the labeling propensity and conditional mean. The paper proves asymptotic normality for their estimator. Under a random-labeling setup with constant selection probability, the authors also show a dominance result over primary-only estimation.

Empirically, the paper evaluates their approach in two settings: a vaccine conjoint setting with noisy discrete LLM labels and a census setting with better-calibrated continuous auxiliary predictions. The paper compares against a relatively small set of baselines of Primary-only, Naive pooling, PPI, and PPI++.

**Compliance With Llm Reviewing Policy:**

Affirmed.

**Final Justification:**

Raised score based on additional experiments. Please include them in the final draft

**Key Questions For Authors:**

Could the authors add comparisons to other relevant work (mentioned above) that use AI-generated outputs as additional information and not necessarily surrogate labels?

**Limitations:**

Yes

**Strengths And Weaknesses:**

Strengths
- The paper shows asymptotic normality of their approach, and shows it dominates a naive labeled-data only baseline

Weaknesses
- Somewhat limited novelty, considering existing prior work [1, 2]. There are several other works on performing valid inference with imperfect forms of synthetic data. Most relatedly, [1] develops a recalibrated version PPI that applies to this setting. This paper (RePPI) formulates the optimal object as the conditional expectation of the loss gradient given observed information, using a ML model, which is similarly show to be better than standard PPI. [2] uses a GMM-based approach to incorporate LLM predictions as additional information into making a more efficient estimator. While the paper tackles a broader setting (with LLM-generated text as well as LLM-generated features), the GMM-based approach can still be applied here.
- The paper also misses a simple baseline of adding in the LLM-predicted features as an additional potentially missing covariate, and applying various techniques from the learning with missing features literature.

[1] Ji, et. al. Predictions as Surrogates: Revisiting Surrogate Outcomes in the Age of AI.
[2] Byun, et. al. Valid Inference with Imperfect Synthetic Data.

---

> ### Author Rebuttal · Authors · 2026-03-29
>
> We thank you for the insightful feedback and for pointing us to closely related work. We agree that the novelty would be clearer with comparisons to the two methods you mentioned, and we have now added both conceptual discussion and empirical benchmarks.
>
> **1. Comparison with RePPI ([1]).**
>
> We agree this is an important benchmark and have added both conceptual and empirical comparisons.
>
> We believe our framework is more general than RePPI on two fronts. (1) We view AI outputs as a *source of information* rather than as a surrogate *label*, which is the perspective taken by RePPI. This allows auxiliary information to be in a **different format from the outcome itself**. For example, in the conjoint experiment, we retain GPT-4o chain-of-thought reasoning text and construct a GAI variant using high-dimensional text embeddings as auxiliary information. RePPI, by design, requires auxiliary information to serve as a surrogate for the outcome and cannot directly use such free-form text. Empirically, both GAI (Label) and GAI (Emb) outperform RePPI in the conjoint setting, and GAI also outperforms RePPI in the census experiment (tables below). (2), our framework handles **more general missing-label probabilities**. In our paper, the labeling propensity e can depend on both X and z, whereas RePPI requires a constant propensity. We have run an additional conjoint experiment with non-constant labeling propensity (reported in our response to Reviewer Zbo8), where GAI continues to outperform all benchmarks.
>
> Importantly, in the special case where the two methods are exactly comparable—the LLM output is used as a label and e is constant across all X and z—RePPI achieves its ideal efficiency by consistently approximating s^* in their paper. We can **theoretically prove that GAI achieves exactly the same asymptotic efficiency**; therefore, our framework incurs no efficiency loss. However, RePPI’s empirical implementation is substantially more complex. It requires a three-fold cross-fitting procedure within the labeled data: the first fold computes an initial estimator θ̂₀, the second fold uses machine learning to estimate the conditional expectation of the loss gradient—a *multidimensional* object—and the third fold computes the optimal weight matrix M̂ and solves the adjusted estimating equation. In contrast, GAI only requires estimating a scalar conditional mean g(X,z) = E[Y|X,z], making it much simpler and more robust in finite samples. In the empirics below, we compare with RePPI. When implementing RePPI, we estimate the recalibrated score on the second fold via L2-regularized logistic regression. Since the available open-source code focuses on linear regression, we implemented the MNL version ourselves following the paper by revising their code. This ease of implementation is critical; it gives rise to better finite-sample performance, which is especially important when labeled data are limited—precisely the regime where AI data augmentation is most valuable.
>
> Comparison to Benchmarks (Conjoint Analysis):
>
> | Method            | MAPE (%) | CI Coverage (%) | Avg CI Width |
> |-------------------|---------:|----------------:|-------------:|
> | Primary           | 23.99    | 92.77           | 1.53         |
> | RePPI             | 33.67    | 78.30           | 1.24         |
> | GMM-Proxy         | 33.36    | 87.50           | 1.22         |
> | GAI               | 16.75    | 95.64           | 1.34         |
> | GAI (Emb)         | 16.42    | 97.36           | 1.47         |
>
> Comparison to Benchmarks (Census Analysis):
>
> | Estimator         | MAPE   | CI Coverage | CI Width  |
> |-------------------|-------:|------------:|----------:|
> | Primary           | 527.43 | 79.20       | 1.16e-05  |
> | RePPI             | 570.27 | 87.60       | 2.02e-05  |
> | GMM-Proxy         | 440.51 | 88.20       | 1.30e-05  |
> | GAI               | 147.56 | 99.60       | 1.54e-05  |
>
> **2. Comparison with Byun et al. ([2]).**
>
> We now include a benchmark based on [2]’s proxy method, reported as GMM-Proxy. The main difference is that our estimator explicitly **corrects for the bias induced by estimating the outcome model from auxiliary information**. We use the AI prediction as a proxy outcome and stack score equations into a joint GMM system solved by two-step efficient GMM (implemented ourselves; no public code). GAI outperforms GMM-Proxy in both studies (tables above).
>
> **3. About the suggested missing-covariate baseline.**
>
> We appreciate this suggestion, but in our setting all covariates X and AI representation z are observed; the missing variable is the human outcome y. So there is no missing-feature problem. That said, in response to your comment and those from other reviewers, we have added many new benchmarks (RePPI, GMM-Proxy, PSPA, Classic-M) and will include all results in the revised paper.
>
> We appreciate your suggestions. They helped us strengthen the paper substantially.

---

> > ### Author Rebuttal · Reviewer_ZoNz · 2026-04-04
> >
> > Thanks! My concerns have been addressed by the additional experiments. Have raised my score.

---

> > > ### Author Response · Authors · 2026-04-05
> > >
> > > We are glad that the additional experiments addressed your concerns, and we will incorporate them into the final version of the paper. Thank you very much for your careful review and thoughtful suggestions!

---

### Decision · Program_Chairs · 2026-04-30

**Decision:**

Accept (regular)

**Comment:**

This paper proposes a framework that treats AI-generated outputs as auxiliary predictive features rather than surrogate labels, to address the problem of valid and efficient causal inference under scarce human-labeled data.

Reviewers are overall positive about this paper and acknowledge that the conceptual reframing of AI outputs as informative features enables principled inference beyond the surrogate-label paradigm (Reviewer ZoNz, Reviewer Zbo8); the empirical results demonstrate consistent improvements in estimation accuracy and confidence interval quality across diverse real-world datasets (Reviewer tk46, Reviewer Zbo8); and the theoretical contributions, including asymptotic normality and a strict variance dominance result under random labeling, are rigorous and well-presented (Reviewer tk46, Reviewer dmw3).

Following the rebuttal, all three reviewers confirmed their concerns were fully or substantially addressed, with Reviewer ZoNz raising their score after the addition of new baselines including RePPI and GMM-Proxy. Reviewer dmw3 maintained their score citing residual novelty concerns, but selected "fully resolved" on the acknowledgement form without identifying specific unaddressed issues.  Therefore, the AC recommends accept for this paper.